*A Nature Portfolio journal*

# NAD$^+$ controls circadian rhythmicity during cardiac aging
Bryce J. Carpenter [1,2,3], Margaux Lecacheur [1,2,3], Yannick N. Mangold [1,3], Kai Cui[1,2,3], Stefan Günther [1,2,3], Marit W. Vermunt[4,5] & Pieterjan Dierickx [1,2,3] ✉

Disruption of the circadian clock as well as reduced NAD$^+$ levels are both hallmarks of aging. While circadian rhythms and NAD$^+$ metabolism have been linked in heart disease, their relationship during cardiac aging is less clear. Here, we show that aging leads to disruption of diurnal gene expression in the heart. Long-term supplementation with the NAD$^+$ precursor nicotinamide riboside (NR) boosts NAD$^+$ levels, reprograms the diurnal transcriptome, and reverses naturally occurring cardiac enlargement in aged female mice. In addition, drastic reduction of NAD$^+$ levels in cardiomyocytes impairs PER2::luc oscillations, which is rescued by NR supplementation. Finally, we demonstrate that changes to the cardiac transcriptome due to NR treatment partially depend on the activity of SIRT1. These findings reveal an essential role for NAD$^+$ in regulation of the cardiac circadian clock upon aging, which opens up new avenues to counteract age-related cardiac disorders.

The circadian clock is an endogenous time-keeping system that allows for the anticipation of rhythmic stimuli such as light/dark and feeding/fasting cycles and is observed in nearly all mammals[1]. The underlying molecular machinery consists of a transcriptional/translational feedback loop including transcription factors such as BMAL1, CLOCK, REV-ERBα/β, and PER2. The clock drives rhythmic mRNA expression of tissue-specific genes both in vivo and in vitro. These oscillations can persist in the absence of external cues but are responsive to changes in the environment, such as light, nutrients and food. In the heart specifically, around 6% of the transcriptome is rhythmically expressed[2], and many clock output genes are implicated in metabolic pathways. Indeed, circadian clocks are tightly linked to metabolic homeostasis, which is best illustrated by genetic disruption of the clock leading to obesity, metabolic syndrome, and cardiovascular defects in rodents[3,4]. In humans, environmental clock disruption (via, for example, shift-work or jet-travel) is also correlated with higher risk for metabolic and cardiovascular disorders[3,5].

Age is one of the greatest risk factors for both likelihood and severity of heart disease, pairing with common pathological parameters such as hypertrophy, reduced cardiac output, and increased oxidative stress[6]. As a highly metabolically active organ, the heart is susceptible to alterations in fuel selection as well as processing, and incidence of diabetes and obesity increases with age[7]. Aging has also been linked to dampening of clock gene expression and circadian processes[8–10], which further associates it with deregulated metabolism and heart disorders[11,12]. Indeed,

whole body knockout (KO) of *Bmal1* in mice leads to premature aging[13] and we and others have previously shown that KO of both *Rev-erbs* in the heart (cardiomyocyte-specific *Rev-erbα/β* double KO (CM-RevDKO)) induces metabolic dysregulation and dilated cardiomyopathy, leading to early lethality[4,14–16]. NAD$^+$, a prominent cofactor and substrate used by hundreds of enzymes[17], has been shown to fall under circadian control in the heart[4,18], and also decreases with age across many species and tissues[11–16]. Generally, alterations in NAD$^+$ levels are largely mediated by either redox status or enzymes that consume NAD$^+$, such as sirtuins, poly ADP-ribose polymerases (PARPs) and CD38[19]. In addition, NAD$^+$ feeds back into the clock via modulating activity of SIRT1, an NAD$^+$-dependent deacetylase working in tandem with the BMAL1:CLOCK dimer that forms the positive arm of the core clock loop[20–22]. This includes regulation of *Nampt* expression[4,20,23], which encodes for the major rate-limiting enzyme in the salvage pathway converting nicotinamide (NAM) into NAD$^+$. This conversion is critical to cardiac NAD$^+$ levels as the heart is incapable of de novo synthesis[24]. Alternatively, cardiac NAD$^+$ can be generated from nicotinamide riboside (NR) by NMRK2, which is upregulated in human and murine heart failure[25]. Often, NAMPT is also decreased in such pathological conditions[24]. A mechanistic study in HEK293 cells has additionally shown that NAD$^+$/SIRT1 is capable of modulating the acetylation status and stability of PER2. Simultaneously, the authors demonstrate that mouse hepatic PER2 levels increase with age but lose transcriptional rhythmicity, in line

[1]Max Planck Institute for Heart and Lung Research, Bad Nauheim, Germany. [2]Cardiopulmonary Institute (CPI), Bad Nauheim, Germany. [3]German Centre for Cardiovascular Research (DZHK), Partner Site Rhine-Main, Bad Nauheim, Germany. [4]Department of Pediatrics, Goethe University Frankfurt, Frankfurt am Main, Germany. [5]German Cancer Consortium (DKTK), Partner Site Frankfurt/Mainz and German Cancer Research Center (DKFZ), Heidelberg, Germany. ✉e-mail: pieterjan.dierickx@mpi-bn.mpg.de

with lowered BMAL1 chromatin occupancy[22]. Therefore, these results suggest that diminished NAD$^+$ due to age may reduce activity of SIRT1, stabilizing PER2 to increase repression of BMAL1, thus dampening oscillations during liver aging. Since high amplitude oscillations are linked to health[26], increasing the amplitude of the core clock and its output might be a viable strategy to reverse aging.

Boosting NAD$^+$ levels via supplementation with NAD$^+$ precursors has proven beneficial in a number of age-associated diseases including murine heart failure models[15,27–30]. However, whether and how NAD$^+$ boosting affects circadian rhythmicity in the heart is not known. In this study, we examine whether increasing NAD$^+$ levels can improve clock function in the heart with age. By using orally-supplemented NR in aging female mice, a treatment strategy we have previously shown to improve cardiac function in disease[15], we identified that transcriptional oscillations of cardiac clock output genes normally affected during aging can be reprogrammed. Concomitantly, NR supplementation reduced age-related heart enlargement and cardiac stress parameters. We verify in vitro that manipulation of NAD$^+$, either by boosting through NR treatment or by loss through pharmacological inhibition of NAMPT, is able to severely impact PER2::luc oscillations in neonatal cardiomyocytes. These results reveal that NAD$^+$ plays an essential role in regulating the diurnal cardiac transcriptome. Lastly, treating a cardiomyocyte cell line with a NAMPT inhibitor, NR, and a SIRT1 inhibitor, we show that part of the transcriptional response to NR is SIRT1-dependent. Together, these data reveal a link between NAD$^+$ metabolism, the circadian clock, and cardiac health during aging, that could be harnessed to ameliorate cardiac defects in aged individuals.

## Results

### Aging alters the diurnal cardiac transcriptome

Aging has been reported to affect circadian gene expression in different tissues[8–10]. To test its effect on diurnal gene expression in the heart, we harvested hearts from young (8 weeks) and old (1 year) female mice at *Zeitgeber* (ZT)22 (night) and ZT10 (day), two timepoints at which the core clock factors *Bmal1* and *Clock* are highly and lowly expressed, respectively, as determined by cardiac qRT-PCR on an independent full circadian female young cohort (Fig. 1A and Supplementary Fig. 1A). We assessed gene expression via bulk mRNA-seq and observed 1,231 diurnally expressed genes in young mice and 302 in old ones (Fig.1B and Supplementary Data 02, adjusted $p < 0.05$). Of those, 142 genes oscillated in both, while 1089 and 160 were uniquely diurnal in young and old mice respectively (Fig. 1B, C). This shows that, in line with other organs[10], fewer genes oscillate in the hearts of older animals. Young-specific diurnal genes were enriched for pathways such as ECM-receptor interaction, focal adhesion, and PI3K-Akt signaling, while oscillators unique to old hearts were enriched for cancer, Ras signaling, and MAPK signaling pathways (Fig. 1D). Genes that displayed diurnal expression in both young and old hearts were enriched for the Circadian Rhythm pathway only (Fig. 1D). Indeed, gene expression of major factors in both the positive and negative arms of the core clock cycle, such as *Clock* and *Per2* respectively, were only mildly affected in aged animals (Fig. 1E). These results suggest that while the core clock does not lose rhythmicity, oscillations of clock-controlled genes are both reduced and rewired in the aging heart.

Since the core clock did not change significantly at the transcriptional level, we wanted to test whether the observed differences of rhythmicity of clock output genes were regulated by alterations of the core clock at the protein level. Therefore, we used *Per2::luciferase* (PER2::luc) reporter mice[31] and isolated cardiomyocytes (CMs) from young (3 months) and old (13 months) female mice. Using an LV200 bioluminescence microscope that can track luciferase signal over multiple consecutive days, we noted that circadian oscillations of old PER2::luc CMs were reduced compared to young CMs in two independent experiments. (Fig. 1F and Supplementary Fig. 1B, C). This suggests that loss of rhythmicity at the protein/functional level of the core clock, rather than transcriptional level, reduces rhythmic clock output with age.

### Aging is associated with cardiac NAD$^+$ decline and hypertrophy which can be reversed by nicotinamide riboside treatment

To investigate the potential link between NAD$^+$ and reduced core clock function in the heart, we evaluated cardiac NAD$^+$ levels during aging. NAD$^+$ levels in old (15 months) female mice were lower compared to young (3 months) mice (Fig. 2A). mRNA levels of genes involved in biosynthesis and consumption of NAD$^+$ were globally unaffected (Fig. 2B). We noted that isolated CMs from old females were significantly larger than CMs from young females (Fig. 2C, Supplementary Fig. 2A), suggesting aging-induced hypertrophic growth. To test whether boosting NAD$^+$ levels can improve diurnal gene expression in the heart, we supplemented the drinking water of 2-month-old female mice with nicotinamide riboside (NR) *ad libitum* for a period of 10 months and harvested animals at the age of 1 year (Fig. 2D). Female mice specifically were chosen due to prior evidence that the cardiac response to NR was most beneficial in females in a model of circadian disruption via KO of the circadian nuclear receptors *Rev-erbα* and *Rev-erbβ* in cardiomyocytes[15]. Upon NR supplementation in females, NAD$^+$ levels in the liver were drastically increased (Supplementary Fig. 2B), whereas hearts only showed a modest trending increase ($p = 0.0551$, Fig. 2E). These findings are in line with previous results[4,32] and could be due to high cardiac metabolic turnover[33]. Nonetheless, while body weight was unaltered (Fig. 2F), we observed that naturally occurring cardiac enlargement was completely abolished in NR-treated mice (Fig. 2G). In line with this, mRNA-sequencing revealed that the cardiac stress marker genes *Anp* and *Bnp* were significantly reduced in older mice given NR (Fig. 2H). As low NAD$^+$ levels in the heart have been associated with hyperacetylation of cardiac proteins[34], we assessed protein acetylation in aged mice. We observed a higher global acetylation status in aged mouse hearts, but NR did not reverse this to a more young-like state (Supplementary Fig. 2C). In conclusion, our data show that NR supplementation reduces aging-induced hypertrophic growth and cardiac stress, thus revealing the therapeutic potential of NAD$^+$ boosting during cardiac aging.

### Supplementation with nicotinamide riboside remodels the diurnal cardiac transcriptome

To investigate whether the clock was affected by NR treatment, we assessed gene expression at ZT22 (night) and ZT10 (day) in old (1 year) NR-treated mice and compared the old, NR treated condition to young and old untreated mice (Fig. 1C). We did not observe major changes in core clock gene expression upon NR (Fig. 3A). From the genes that were rhythmic in both young and old mice (Fig. 1C, n = 142), the majority (n = 86/142) were unaffected by NR and stayed rhythmic, while 56/142 genes lost rhythmicity (Fig. 3B, C). From the genes that lost rhythmicity upon aging (Fig. 1C, n = 1089), most were unaltered and still lost rhythmicity (n = 979), but 110 remained diurnally expressed upon NR. From the latter, 92 kept their normal peak and trough (based on 2 ZTs), but 18 inverted their rhythmic expression pattern. The genes that gained rhythmicity with age (Fig. 1C, n = 160), predominantly lost this pattern upon NR (n = 111/160). Next to the previously described genes (Fig. 1C), we identified 213 genes that were normally not rhythmic in young or old but gained de novo rhythmicity during aging upon NR treatment (Fig. 3B). KEGG pathway and epigenetic Landscape In Silico deletion Analysis (LISA) revealed that genes that were rhythmic in young and old, but lost diurnal expression following NR, are enriched for longevity regulating and neurotrophin signaling pathways, with NKX2-5, TBX3, and NR3C1 as putative transcriptional regulators of these genes (Supplementary Data 02, $p < 0.01$). Transcripts that lost rhythmicity with age but remained diurnally rhythmic upon NR encompassed FoxO, AMPK, and adipocytokine signaling and are potentially regulated by TBX3, NKX2-5, and JARID2. Genes not rhythmic in both young and old hearts, but with de novo oscillatory expression upon NR treatment were primarily involved in Apelin signaling, choline metabolism in cancer, and sphingolipid signaling pathways and are potentially regulated by TBX3, MEF2A, and SUZ12 (Supplementary Data 02). These results suggest that NR reprograms the diurnal transcriptome in the aging heart and that different TFs contribute to distinct types of reprogramming.

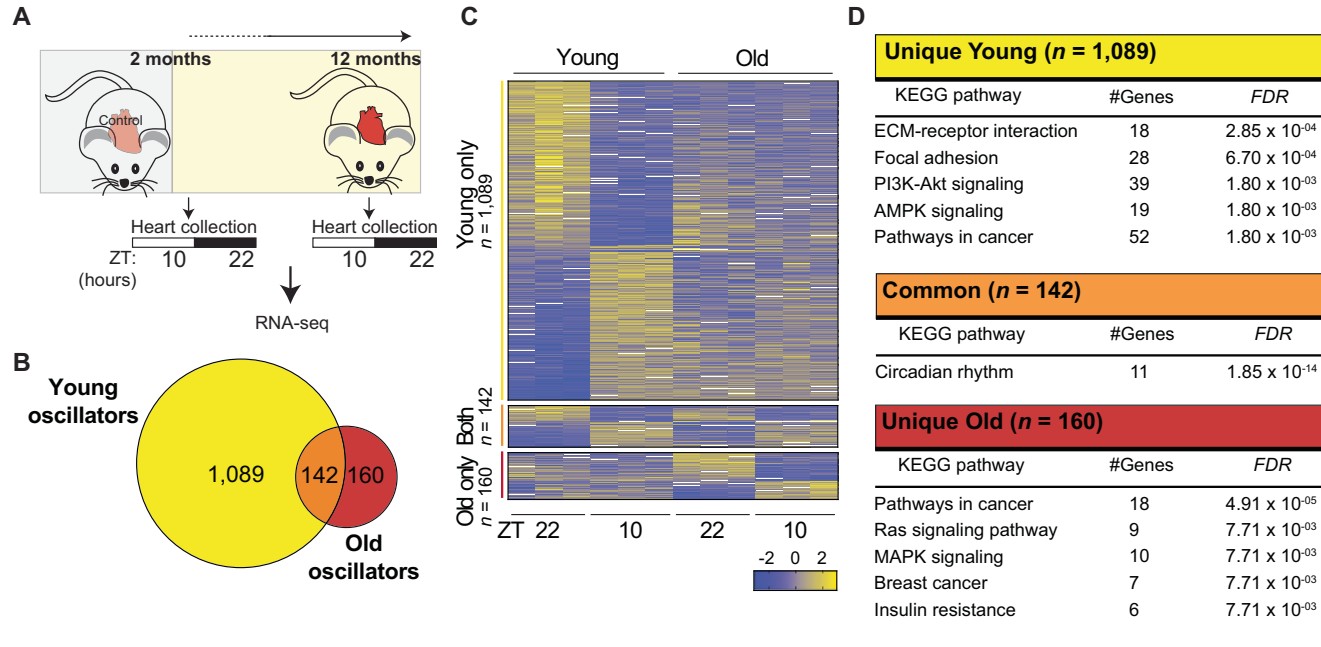

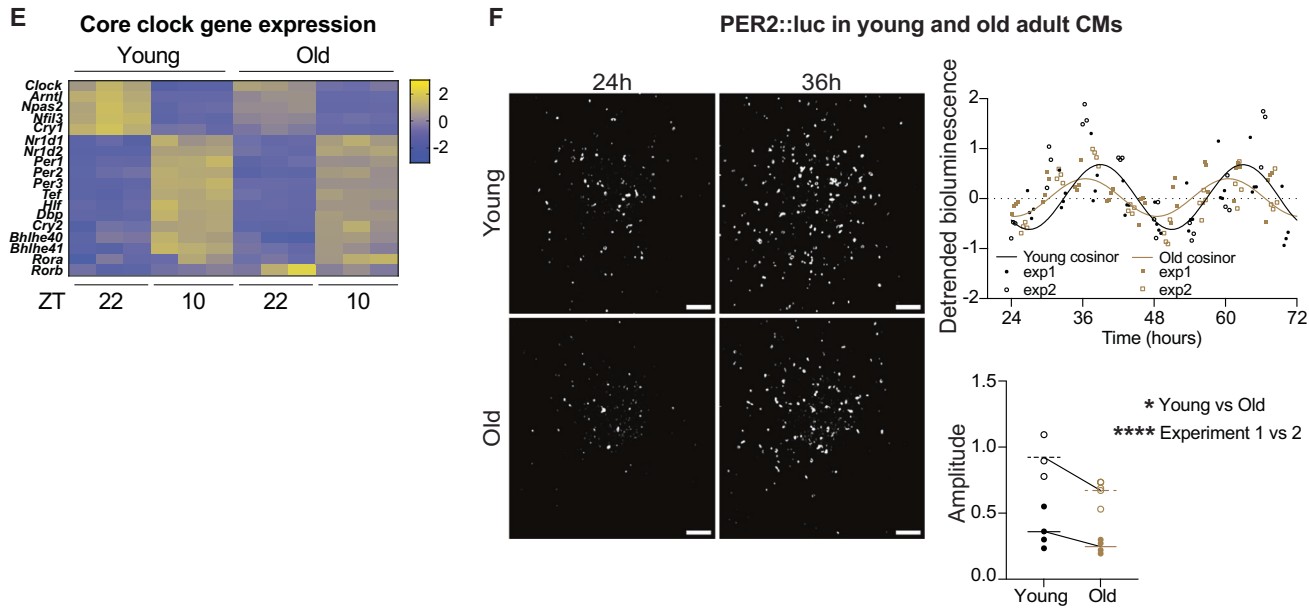

**Fig. 1 | The cardiac diurnal clock is reprogrammed during aging. A** Schematic representation of experimental setup in female mice (adapted from a public domain image available on clcker.com). **B** Venn diagram showing the overlap between oscillators in young (2 months) and old (1 year) female hearts ($n = 3$/condition, DESeq2 ZT10 vs ZT22, adjusted $p < 0.05$). **C** Heatmap depicting z-normalized expression levels for genes in (**B**). **D** KEGG pathway analysis on all DEGs from (**B, C**). The analysis was performed via the use of Enrichr[52]. **E** Heatmap depicting z-normalized expression levels for core clock genes in young (2 months) and old (1 year) female hearts. **F** Representative bioluminescent tracks from cardiomyocytes isolated from young and old female Per2::luc animals and their detrended quantification ($n = 3$ wells per age per experiment, data from two independent experiments separately demarcated). Circadian oscillations were analyzed using the COSINOR algorithm[53] to determine amplitude of rhythmicity by well. Graphs plot individual wells, with lines representing cosinor curve fit to merged experiments or average calculated amplitude per age and experiment. *$p < 0.05$; ****$p < 0.0001$; by Two-way ANOVA.

Specifically, NR treatment retains a fraction of young-specific oscillatory genes in old mice, but also prevents development of rhythmicity in a majority of old-specific oscillators. Together with our data on NAD$^+$ loss and cardiac hypertrophy with age which can be reversed by NR, this suggests that NAD$^+$ levels are an essential regulator of transcriptional rhythmicity and cardiac health.

## NAD$^+$ modulation affects PER2 rhythmicity

The fact that regulation of clock maintenance and function during aging relies on NAD$^+$ levels has not previously been shown in a cardiac context. To

further validate this finding and better isolate variables from in vivo aging, we sought to use a model in which we can control NAD$^+$ levels more directly and thus correlate introduced NAD$^+$ alterations to clock readouts over several days. Therefore, we picked neonatal cardiomyocytes from PER2::luc mice as a model to avoid the rapid morphological changes associated with culture of adult cardiomyocytes[35] and to be able to monitor rhythmicity over extended treatment times. Neonatal cardiomyocytes were treated with FK866, a potent NAMPT inhibitor, to lower NAD$^+$ levels as a proxy for the aged in vivo condition. After two days of FK866 treatment, NAD$^+$ levels were dramatically reduced (Fig. 4A). Concurrent treatment with FK866 and

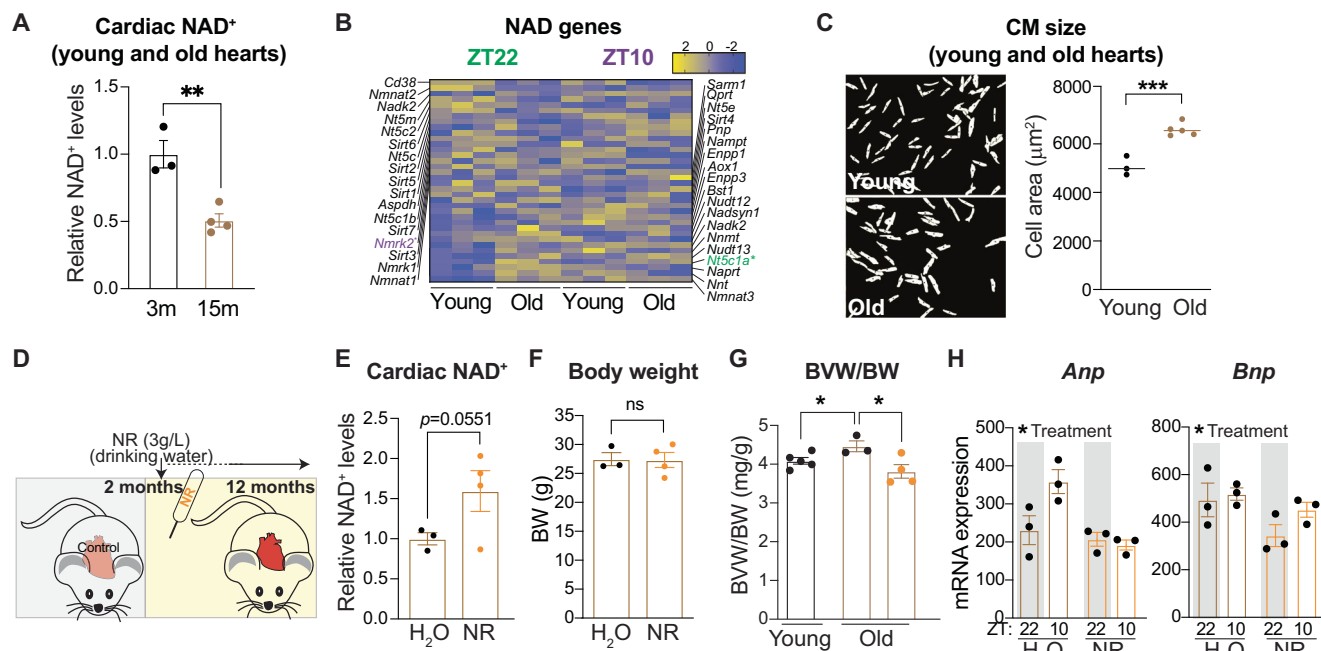

**Fig. 2 | Cardiac NAD⁺ decreases with aging, with restoration by supplementation with nicotinamide riboside. A** Cardiac NAD⁺ levels from young (3 months(m)) and old (15 months(m)) female mouse hearts ($n = 3$ for young, $n = 4$ for old) at ZT22. **B** Heatmap depicting z-normalized expression levels for NAD-related genes in young (2 months) and old (1 year) female hearts at ZT22 and ZT10. Genes indicated in green and magenta text are differentially expressed at ZT22 and ZT10 respectively. **C** Cellular area from adult cardiomyocytes isolated from female young (3 months) and old (15 months) hearts ($n = 3$ young mice, $n = 5$ old mice). **D** Schematic representation (adapted from a public domain image available on clckr.com) of NAD⁺ boosting strategy in female mice with NR. **E** Cardiac NAD⁺ concentration and **F** body weight (BW) of female mice treated with NR versus H₂O (control) for 10 months starting at an age of 2 months ($n = 3$ control, $n = 4$ NR-treated) at ZT10. **G** Biventricular heart to body weight (BVW/BW) ratio in young and old mice treated with NR versus H₂O (control) ($n = 5$ young mice, $n = 3$ old mice, $n = 4$ old mice with NR). **H** *Anp* (*Nppa*) and *Bnp* (*Nppb*) mRNA expression (DESeq2 normalized counts) in hearts from NR vs control treated mice ($n = 3$/ condition) at ZT10. ns non-significant; *$p < 0.05$; **$p < 0.01$; ***$p < 0.001$; by 2-tailed unpaired Student's $t$ test for (**A**), (**C**), (**F**), and (**G**): by 1-tailed unpaired Student's $t$ test for (**E**); by Two-way ANOVA for (**H**).

NR boosted NAD⁺ levels (Fig. 4A). These effects persisted through 6 days of treatment, eventually altering NADH levels as well. Neither treatment had apparent effects on cell morphology within this timeframe (Supplementary Fig. 3). In line with lowered NAD⁺, we noted a drastic reduction in PER2::luc rhythms within four days of FK866 treatment, indicating an impairment of the central circadian clock (Fig. 4B). This reduction in rhythmicity could be completely prevented by concomitant addition of NR which restored NAD⁺ levels (Fig. 4B). In addition, adding NR after cells were initially treated with FK866 alone enhanced rhythmicity again (Fig. 4B). This suggests that NR can be used to prevent as well as to treat reduction of rhythmicity. These results show that NAD⁺ levels correlate with PER2 protein rhythms in cardiomyocytes and confirm that NR can improve PER2 rhythms. Therefore, transcriptional changes to clock-controlled genes induced by NR in vivo are likely to depend on core clock protein rhythmicity regulation by NAD⁺ levels.

## SIRT1 activity mediates transcriptional response to NR

Studies in the liver have revealed mechanisms by which NAD⁺ contributes to core clock complex stability and output via SIRT1, an NAD⁺-dependent deacetylase. SIRT1 regulates PER2 localization and stability via targeted deacetylation[21,22]. Similarly, SIRT1 has been previously shown to interact with CLOCK and influence the acetylation status of BMAL1, thereby also directly modulating the positive arm of circadian regulation and rhythmicity of output genes[36]. Given our results that indicate multiple connections between NAD⁺ and downstream rhythmicity in the heart, including PER2::luc protein rhythmicity in both neonatal and adult cardiomyocytes, we hypothesized that SIRT1 may contribute to the response to NR in our cardiac experiments too. To evaluate whether NAD⁺-induced reprogramming requires SIRT1 in the cardiac context, we performed mRNA-seq on HL-1 cells, an adult CM cell line[37], that were untreated or treated with FK866

only or FK866 and NR in the presence or absence of Ex527, a potent inhibitor of SIRT1 activity. Like in neonatal cardiomyocytes, FK866 was used to induce a low NAD⁺ state and thus mimic the reduced NAD⁺ levels in aged mice (Fig. 5A). Concomitant treatment with FK866 and NR, in the absence or presence of Ex527, boosted NAD⁺ levels relative to treatment with only FK866 (Fig. 5A). Comparison of untreated and FK866 alone (low NAD⁺) cells resulted in differential expression of 643 genes (adjusted $p < 0.01$, $\log_2 FC > 0.58496$) (Fig. 5B and Supplementary Fig. 4A). After simultaneous treatment with both FK866 and NR, 619 out of these 643 were no longer differentially expressed compared to untreated cells, demonstrating that the majority of FK866 response genes are NAD⁺ dependent and that FK866 treatment is a good proxy for low/aged NAD⁺ cell states. To select genes that are affected by NR in a reduced NAD⁺ state, we directly compared FK866 alone (low NAD⁺) and FK866 + NR (NAD⁺ boosted). 618 NR response genes were identified (Fig. 5B, C and Supplementary Fig. 4A). Of note, *Sirt1* did not meet fold-change cutoffs, and also was not significantly different in the in vivo data (Supplementary Fig. 4B). We next examined whether the NR-response genes still changed if SIRT1 activity was inhibited. In the presence of Ex527, the majority (468/618 genes) of the transcriptional NR response genes were no longer DE, indicating they require intact sirtuin activity (Fig. 5B, C and Supplementary Data 03). The remainder of the genes (150/618) demonstrate, however, that SIRT1 is not the sole mediator of the NR response. Nonetheless, in line with previous observations in liver[22], NAD⁺ mediates transcription predominantly in a SIRT1-dependent manner in HL-1 cardiomyocytes.

## Discussion

NAD⁺ metabolism, aging and the circadian clock are interconnected. Here, we show that rhythmicity of clock output genes is reduced and rewired with age in the heart and that long-term NR supplementation to increase NAD⁺

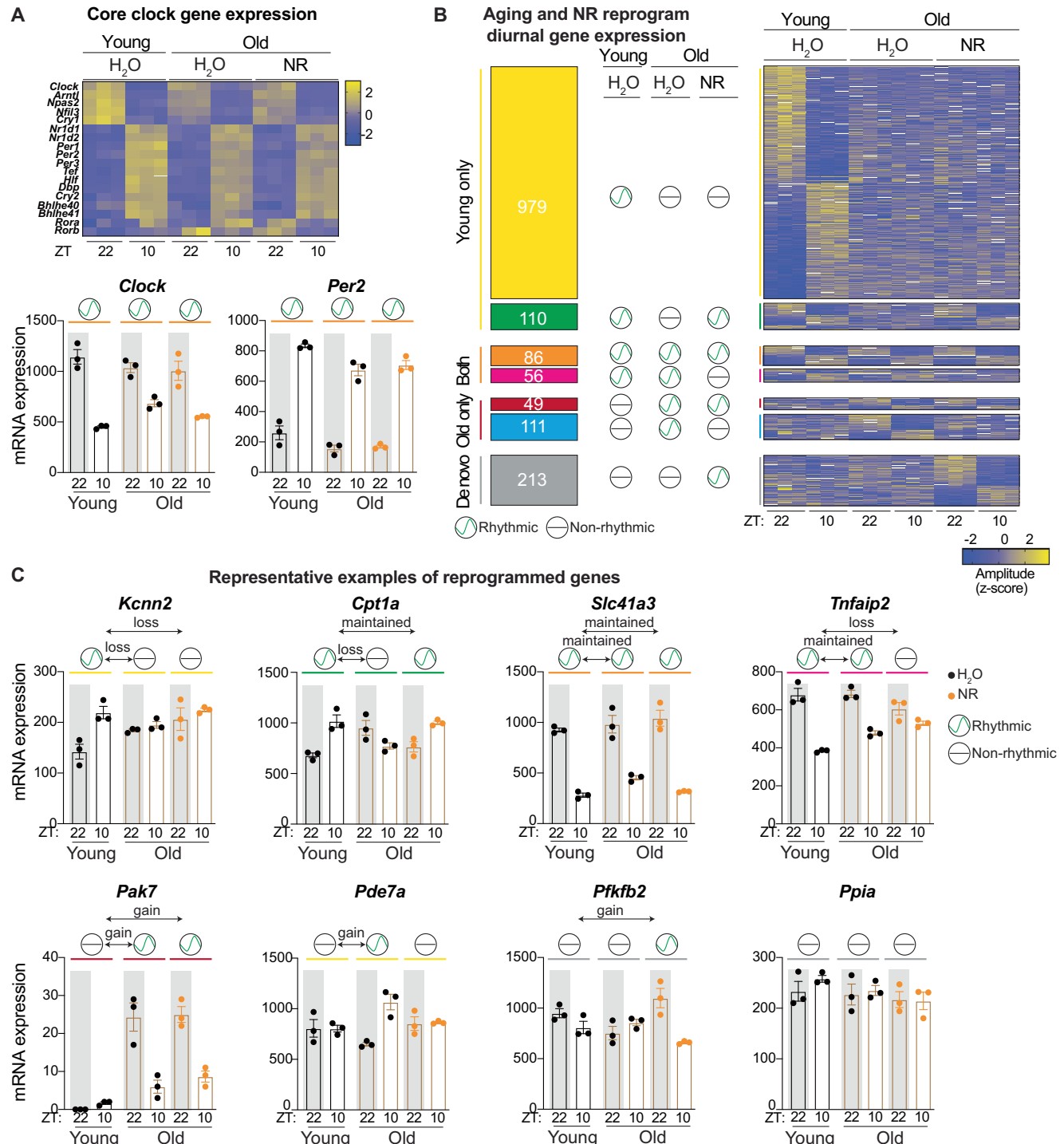

**Fig. 3 | Nicotinamide riboside supplementation reprograms the cardiac diurnal transcriptome. A** Heatmap depicting z-normalized expression levels for core clock genes in young and old NR vs control treated female hearts. **B** Identified categories of oscillators and heatmap depicting their z-normalized expression values, depending on their diurnal expression in each condition (DESeq2 ZT10 vs ZT22, adjusted $p < 0.05$, $n = 3$/condition). **C** Gene expression (DESeq2 normalized counts) for representative genes of each class. Graphs show mean ± s.e.m. Oscillation sign depicts differential expression (adjusted $p < 0.05$ assessed by DESeq2, $n = 3$/condition).

levels can partially restore diurnal oscillation of young-specific clock output genes. This occurs without NR supplementation altering transcription of the core clock itself. However, our data suggest a dependence on core clock changes at the level of protein functionality, which is partially mediated by SIRT1 activity. While proper amplitudes of circadian rhythms are clearly associated with health[26], the importance of rhythmicity in downstream outputs is often not obvious in terms of delineating its role in standard physiology, compensation, or pathology. However, advanced aging is

generally a less healthy state. We show that many (young-specific) genes lose rhythmicity with age, but also that genes associated with cancer and insulin resistance start oscillating specifically in old hearts. Our findings that NR treatment results in maintenance of rhythmicity of otherwise young-specific oscillators (110 genes) and prevent age-induced rhythmicity of a subset of old-specific oscillators (111 genes) is encouraging. Interestingly, the genes that remain rhythmic during aging upon NR treatment are linked to FOXO signaling, a pathway critical to heart development and aging. Since SIRT1

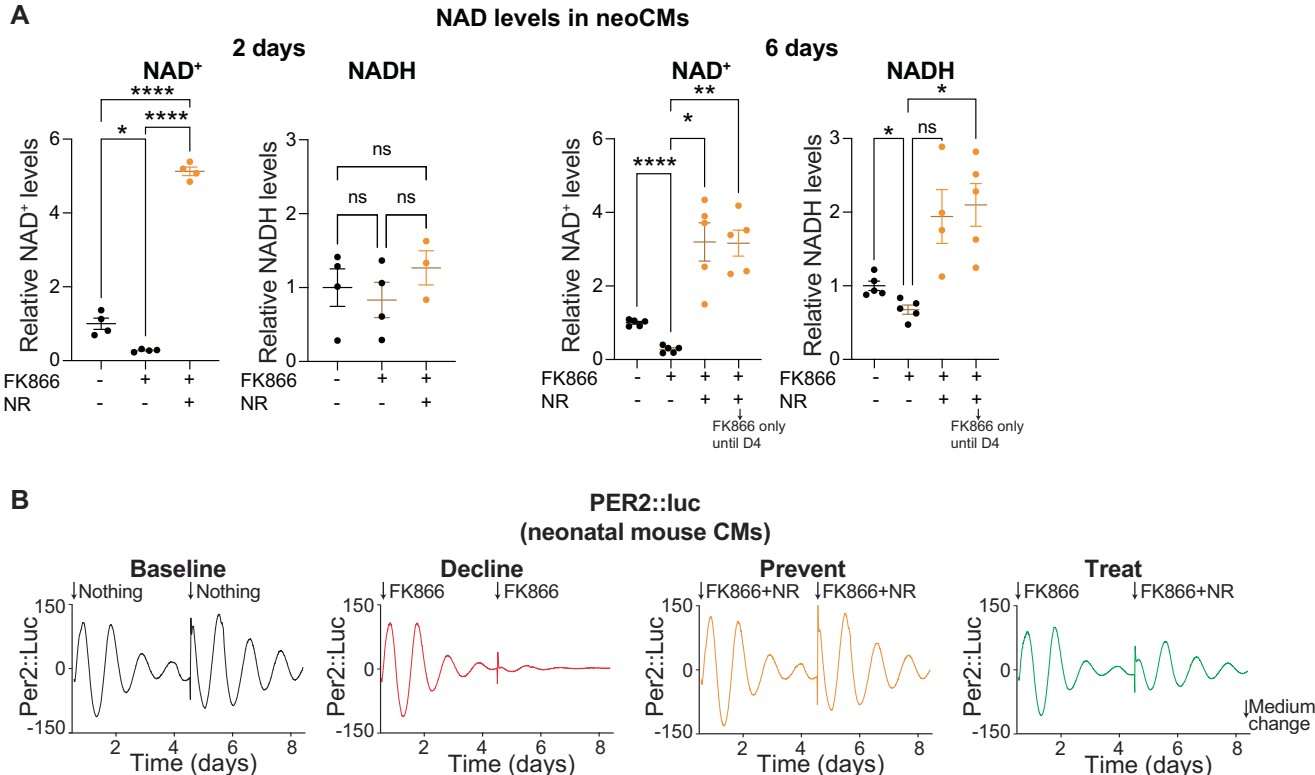

**Fig. 4 | Modulation of NAD⁺ impacts oscillations of PER2::Luc protein level.**
**A** NAD⁺ and NADH levels of neonatal mouse cardiomyocytes after 2 or 6 days of treatment with NAMPT inhibitor FK866 (100 nM) with or without NAD precursor NR (500 µM) ($n = 4$/condition, except $n = 3$ for NR + FK866 NADH, at day 2; $n = 5$/condition, except $n = 4$ for NR + FK866 NADH, at day 6).
**B** Average background-subtracted PER2::luciferase oscillations from neonatal mouse cardiomyocytes treated with same conditions in (**A**). Cells were derived from *Per2::luciferase* mice and monitored ex vivo in a Lumicycle ($n = 4$). Arrows indicate medium change at day 4 and addition of fresh FK866 and NR, including treatment transition of a subset of FK866-only cells into FK866 + NR. Graphs in (**A**) show mean ± s.e.m. ns non-significant; *$p < 0.05$; **$p < 0.01$; ***$p < 0.001$; ****$p < 0.0001$; by Brown-Forsythe ANOVA with Dunnett's T3 multiple comparisons test on displayed pairs for (**A**).

can directly affect the transcriptional activity of several FOXO proteins via NAD⁺-dependent deacetylation[38,39], it might be that maintenance of rhythmicity in NR-treated aged mice is partially facilitated by increased SIRT1 activity on FOXO proteins. LISA, a tool which predicts upstream transcriptional regulators, forecasts these changes could be mediated by TBX3 and JARID2, both factors critical to cardiac development[40,41]. Genes that did not oscillate in young and old, but only started oscillating in NR-treated old mice were found by LISA to correlate with hypertrophy factors like MEF2A[42] and SUZ12[43]. Further investigation into how these factors alter rhythmicity and how they affect the linked pathways will be necessary to fully reveal the mechanisms by which NR treatment protects against cardiac aging and cardiovascular disease. Importantly, we here show that NR treatment was safe in female mice and prevented naturally occurring cardiac enlargement observed with age. These effects should continue to be explored across further aging and between genders, as gene expression programs depend on these parameters[44]. Likewise, the development of either robust cardiac aging models in vitro or improved technologies for acute modulation of NAD⁺ metabolism and continuous readout in vivo would greatly benefit translational relevance; the current study was limited in part by the challenges of heart-targeted pharmacological intervention on mice and primary cardiac cells, necessitating the use of models such as the HL-1 cell line at present. Nevertheless, we have here demonstrated that: 1) even limited aging alters rhythmic gene expression and cardiac NAD⁺ levels in vivo; 2) that these changes can be partially reversed by NR treatment; 3) that manipulation of cardiomyocyte NAD⁺ is possible via small molecules and affects PER2 rhythmicity; and 4) that these changes are partially dependent on the activity of SIRT1. This provides evidence to support the known but understudied involvement of NAD⁺ in cardiac clock

rhythmicity and aging, possibly through a loop similar to the NAD⁺-SIRT1-PER2 axis discovered in the liver[21,22]. Cardiac *Sirt1* mRNA levels were not different between young and old mice, but reduced protein or enzymatic activity of SIRT1 due to lower NAD⁺ levels in old mice might contribute to clock output remodeling. In line with SIRT1 being a protein deacetylase, we observed increased global cardiac protein acetylation in aged mice. However, we were unable to confirm PER2 being among the hyperacetylated proteins. NR did not revert increased acetylation to a young-like state, but global acetylation is a reading of combined activity of several other sirtuins, many more (de)acetylases, as well as cellular processes. Therefore, NR-responsive genes may be downstream of relatively few and specific factors with altered acetylation upon aging and/or after NR treatment. Investigation of additional NAD⁺-dependent processes on the cardiac clock (output) such as redox status and NAD⁺ consumption (by PARPs, for example) may reveal cooperative or alternative methods of NAD⁺-mediated rhythmicity. Indeed, the regulation of NAD⁺ generation and the clock are both multivariate and tissue-specific. Basse et al. recently demonstrated that the core clock in brown adipose tissue is highly dependent on NAD⁺ biosynthesis, while the skeletal muscle clock is largely refractory to NAMPT depletion[45]. We show a clear effect of NAD⁺ modulation on the cardiac clock output, but to what extent the cardiac circadian clock depends on NAMPT-mediated NAD⁺ biosynthesis as a function of age needs to be tested. Since NAD⁺ supplementation strategies are regularly proposed to combat aging and age-related diseases (e.g. heart failure), understanding the effects of NAD⁺ on the cardiac clock will be crucial. Our findings reveal a clear link between NAD⁺ and cardiac diurnal rhythms and offer potential avenues to treat and prevent cardiovascular diseases appearing with age by using NR to partially restore a young-like oscillatory program.

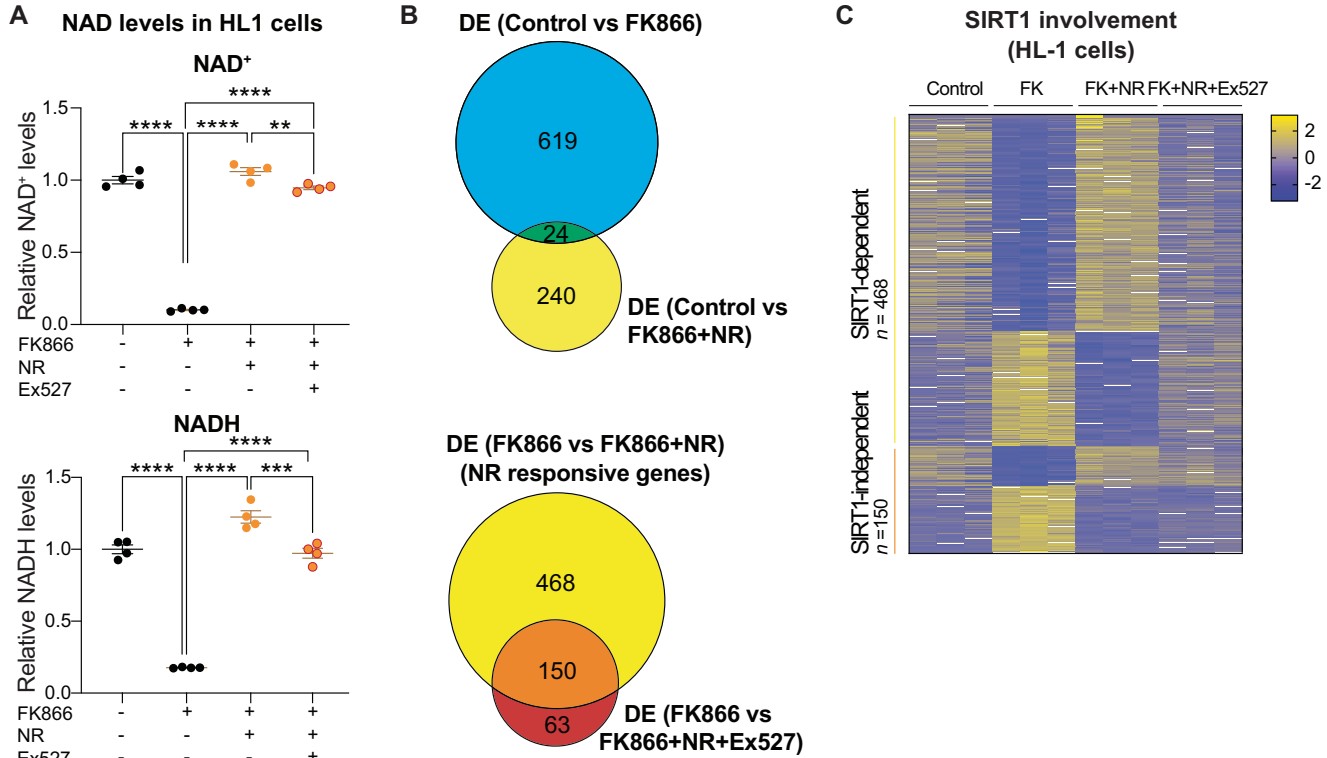

**Fig. 5 | The transcriptional response to NR largely depends on intact SIRT1 activity. A** NAD$^+$ levels of HL-1 cells untreated, treated with 5 nM FK866 (FK), 5 nM FK866 + 100 µM NR, or 5 nM FK866 + 100 µM NR + 20 µM Ex527 ($n = 4$/condition). **B** Venn diagrams of genes differentially expressed (DESeq2, adjusted $p < 0.01$, log$_2$FC > 0.58496) between untreated cells and cells treated with either

FK866 or FK866 + NR or cells treated with FK866 and cells treated with either NR or NR+Ex527 ($n = 3$/condition). **C** Heatmap highlighting NR response genes, organized by whether or not differential expression was maintained with SIRT1 inhibition. Graphs show mean ± s.e.m. **$p < 0.01$; ***$p < 0.001$; ****$p < 0.0001$ by One-way ANOVA.

## Methods

### Animals

Female mice (*mus musculus*) on a mixed C57Bl6J/N strain, representing diversified backgrounds used in previous NR treatments, at the age of 2–4 months ("young" condition) or 12–15 months ("old" condition) were used for all experiments unless stated otherwise in figures and figure legends. They were housed under 12 h light/12 h dark conditions and fed a standard chow diet *ad libitum* with free access to water, with weekly cage changes and daily observation checks. NR (Elysium Health, NY, United States) was added to the drinking water (3 g/L) of randomized cages of 3–5 mice in light-protected bottles that were changed twice per week, starting at the age of 2 months. Animal experiments for RNA-seq were performed in the lab of Prof. Dr. Lazar. We have complied with all relevant ethical regulations for animal use per the guidelines of the Institutional Animal Care and Use Committee (IACUC) of the University of Pennsylvania and by the responsible Committee for Animal Rights Protection of the State of Hessen (Regierungspraesidium Darmstadt, Wilhelminenstr. 1-3, 64283 Darmstadt, Germany) under project number B2-2018.

### Cell culture

All primary cells and cell lines were maintained in humidified incubators at 37 °C and 5% CO$_2$. All media were filtered prior to use and medium tested at every medium change for mycoplasma. Primary cells were provided medium as described below in their respective isolation protocols. The HL-1 Cardiac Muscle Cell Line (Sigma-Aldrich, SCC065), originally derived from an atrial tumor of an adult female C57BL/6J mouse, was obtained directly from Sigma-Aldrich. After receipt, HL-1 cells were expanded over twelve passages at a 1:3 ratio until experimental use. HL-1 cells were kept on gelatin-coated plates and in Claycomb medium (Sigma-Aldrich, 51800C) supplemented with 10% FBS (Capricorn Scientific, FBS-12A), 0.1 mM norepinephrine (Sigma-Aldrich, A0937), 2mM L-Glutamine (Sigma-

Aldrich, TMS-002-C) and 1× penicillin/streptomycin (Sigma-Aldrich, TMS-AB2-C), stored light-protected. Cells were always plated at 125,000 cells/mL and allowed to grow to 90% confluency before splitting. The same medium was used for experimental treatments with the addition of 5 nM FK866, 5 nM FK866 + 100 µM NR, or 5 nM FK866 + 100 µM NR + 20 µM Ex527 immediately after passaging.

### Adult mouse cardiomyocyte isolation

Cardiomyocytes from adult female animals were collected via the Langendorff-free perfusion method[35] with minor modifications. All buffers were prewarmed to 37 °C and a heating pad placed under dishes and syringes to maintain temperature during the protocol. Mice were euthanized by cervical dislocation, followed by opening of the chest cavity and immediate flushing of the heart right ventricle with 7 mL of EDTA buffer. Then, the aorta was clamped shut with hemostatic forceps and cut free from the mouse and the entire heart transferred to a dish containing EDTA buffer. The left ventricle was slowly perfused for 6 min with more EDTA buffer, with visible clearance of blood from all chambers. The EDTA buffer was replaced after moving the heart to a new dish of perfusion buffer by a short chase of 2 min of perfusion buffer. The heart was then moved to a filled dish of collagenase buffer and perfused with 20–60 mL of collagenase buffer until the heart took on a fuzzier, paler appearance, alongside development of additional perforations and effluent cardiomyocytes. Finally, the heart was put into a smaller (~5 mL) volume of clean collagenase buffer, mechanically separated by manual use of forceps, followed by gentle trituration for 2 min with a 1000 µL pipette and a wide-bore tip. The full volume was transferred to a 15 mL tube through a 100 µm filter and spun down for 2 min at 20 g. The supernatant was removed, and the resulting cell pellet equilibrated to full calcium concentration of the culture medium in three steps; each step consisting of 15 min of cell settling by gravity and replacement of the supernatant with progressively higher concentrations of culture medium

diluted with perfusion buffer (1 part medium: 3 parts perfusion buffer; 1:1; 3:1). The final cell pellet was resuspended in 2 mL pre-warmed culture medium, 10 μL added to each side of a Neubauer cell counter, and all eight corner squares counted for both rod-shaped and rounded cardiomyocytes to assess viability, followed by dilution to target concentration of 50,000 rod-shaped cells per mL and plating at 5000 rod-shaped cells per well of a 96-well plate. Medium was changed gently after the first three hours and then cells were taken for imaging.

## Microscopic real-time bioluminescence analysis

Bioluminescence from adult PER2::luc cardiomyocytes was assessed with an LV200 microscope (Olympus) in a humidified chamber under 5% $CO_2$, at 37 °C guided by a controller (Tokahit), using a 10×0.3NA Plan Semi Apo objective (Olympus). Cells were synchronized with 100 nM dexamethasone for 2 h and changed to adult cardiomyocyte culture medium, containing 1 mM D-Luciferin Potassium Salt (Promega). A transmitted light image was recorded prior to the start of each imaging run to localize the cells. Bioluminescence was detected for multiple consecutive days, using an EM-CCD camera (Hamamatsu), with exposure times of 8 min, and an interval of 6 h. Subsequent bioluminescence values were detrended per condition by subtraction of a second-degree polynomial line of best fit set to the entire window of collection, thus adjusting for changes in background signal over the imaging period. Image series were analyzed in ImageJ.

## Neonatal mouse cardiomyocyte isolation

Neonatal mouse cardiomyocytes from female P0-P3 *PER2::Luc* animals were harvested with a primary mouse cardiomyocyte isolation kit (Pierce) following the manufacturer's protocol. Neonatal mouse hearts were excised and immediately placed in separate tubes with 500 μL of ice-cold Hanks' Balanced Salt Solution (HBSS), then minced with clean surgical scissors before two additional washes with ice-cold HBSS. The minced hearts were then incubated in the provided enzyme solution, incubated at 37 °C for 35 min, washed twice with cold HBSS, and finally pipetted 30 times to break tissue in 500 μL completed medium. Final cell suspensions were combined and diluted with completed culture medium to desired concentration of ~400,000 cells/mL and plated according to plate size.

## Bioluminescent recording and data analysis

Neonatal mouse cardiomyocytes were synchronized with 100 nM Dexamethasone (Sigma) for 2 h in standard culture medium from the isolation kit. Subsequently, medium was changed to recording medium (Powdered DMEM (Corning), 10 mM HEPES, 1% P/S, 5% FBS, 0.035% Sodium bicarbonate, 100 μM D-Luciferin Potassium Salt (Promega)) with the addition of 100 nM FK866, FK866 + 500 μM NR, or nothing. Culture dishes were sealed with high vacuum grease (Dow Corning) and monitored using a LumiCycle32 device (Actimetrics) at 37 °C. After 4 days, recording medium was refreshed with the addition of fresh FK866, FK866 + NR, or nothing. Bioluminescence from each dish was continuously recorded (integrated signal of 70 s with intervals of 10 min). Raw data (counts/s) were baseline subtracted.

## NAD+ measurements on tissue

Hearts were harvested and snap frozen in liquid nitrogen. Hearts were powdered and subsequently lysed in 0.6 M perchloric acid. Cardiac $NAD^+$ levels were measured by a cycling enzymatic assay as previously described[4]. In short, ~20 mg of powdered heart tissue was homogenized in 400 μL of ice-cold 0.6 M perchloric acid, then centrifuged at max speed for 15 min at 4 °C to pellet debris. An aliquot of the resulting supernatant was then diluted 1:50 with 100 mM sodium phosphate buffer to form a working concentration. 5 μL of working sample or 5 μL of standards (six 1:1 serial dilution steps from 1 μM NAD and a 0.1 M sodium phosphate buffer blank) was then combined with 95 μL of cycling mix (0.1 M sodium phosphate, 0.1 mg/mL bovine serum albumin, 10 mM nicotinamide, 2% ethanol, 10 μM flavin mononucleotide, 20 μM resazurin, 110 μg/mL alcohol dehydrogenase, 110 μg/mL diaphorase). A Tecan Infinite M200 Pro microplate reader with fluorescence

excitation at 544 nm and emission at 590 nm was used to measure immediately after cycling mix addition and after 20 min of incubation. The concentration of $NAD^+$ was then determined by subtracting the prior measurement from the latter for every well, then plotting the subsequent value on the standard curve. $NAD^+$ values were normalized to tissue weight.

## NAD+ measurements on cells

Measurements were performed using the Promega NAD/NADH Glo-Assay kit according to manufacturer instructions. In short, cells were lysed in-well with 50 μL 0.2 M NaOH + 1%DTAB and diluted 1:1 with dPBS. 25 μL was then transferred to a new tube on ice, and remaining lysate snap frozen. Working lysate was then treated with 12.5 μL 0.4 M HCl at 60 °C for 15 min, and allowed to rest at RT for 10 min before addition of 12.5 μL of 0.5 M Trizma Base. 25 μL of resulting mixture was then immediately combined 1:1 with complete assay reagent mix and incubated for 30 min before luminescence measurement at plate reader, followed by normalization to protein concentration per sample (as obtained by Pierce BCA quantification on the separate snap-frozen lysate).

## Cardiomyocyte size assessment

Images of adult CMs were taken with an EVOS microscope at 10× magnification, 3 h after plating. Size of rod-shaped CMs was analyzed using Cellpose software[46] to segment the cells in the images first, Ilastik[47] was used to classify round or rod-shaped cells and Fiji was used for object size measurement.

## RNA isolation, quantitative RT-PCR, mRNA-sequencing, and analysis

Total RNA was extracted from heart tissues (TRIZOL) or HL-1 (RLT) using RNAeasy (Qiagen) according to the manufacturer's instructions, and treated with DNase (Qiagen) before library preparation and cDNA synthesis. cDNA was generated using High-Capacity cDNA Reverse Transcription Kit (Applied Biosystems). Quantitative PCR reactions were performed using PowerSYBR Green PCR Master Mix (Applied Biosystems) with specific primers on a QuantStudio 6 Flex instrument (Applied Biosystems). mRNA expression was normalized to the housekeeping gene *Ppib* for all samples. Primer sequences for qRT–PCR: *Ppib-fw*, 5'-GCAAGTTC-CATCGTGTCATCAAG-3'; *Ppib-rv*, 5'-CCATAGATGCTCTTTCCTCC TG-3'; *Bmal1-fw*, 5'-TAGGATGTGACCGAGGGAAG-3'; *Bmal1-rv*, 5'-TCAAACAAGCTCTGGCCAAT-3'; *Clock-fw*, 5'-AAAGACGGCGA-GAACTTGG-3'; *Clock-rv*, 5'-GGAGGCAGAAGGAGTTGGG-3'. mRNA-seq of in vivo mouse heart samples was performed by our in-house RNA-sequencing core. RNA and library preparation integrity were verified with LabChip Gx Touch 24 (Perkin Elmer). 1 μg of total RNA was used as input for VAHTS Stranded mRNA-seq V6 Library preparation following manufacture's protocol (Vazyme). Sequencing was performed on NextSeq2000 instrument (Illumina) with 1x72bp single end setup. Trimmomatic version 0.39 was employed to trim reads after a quality drop below a mean of Q15 in a window of 5 nucleotides and keeping only filtered reads longer than 15 nucleotides (Trimmomatic[48]: a flexible trimmer for Illumina sequence data). Reads were aligned versus Ensembl mouse genome version mm10 (Ensembl release 101) with STAR 2.7.10a (STAR[49]: ultrafast universal RNA-seq aligner). Alignments were filtered to remove: duplicates with Picard 3.0.0 (Picard: A set of tools (in Java) for working with next generation sequencing data in the BAM format), multi-mapping, ribosomal, or mitochondrial reads. Gene counts were established with featureCounts 2.0.4 by aggregating reads overlapping exons excluding those overlapping multiple genes (featureCounts[50]: an efficient general-purpose program for assigning sequence reads to genomic features). The raw count matrix was normalized with DESeq2[51] version 1.36.0 (Moderated estimation of fold change and dispersion for RNA-Seq data with DESeq2). Contrasts were created with DESeq2 based on the raw count matrix. Genes were classified as significantly differentially expressed at average count >5, multiple testing adjusted $p < 0.05$. The Ensemble annotation was enriched with UniProt data (Activities at the Universal Protein Resource (UniProt)). KEGG pathway

enrichment was assessed via Enrichr[52]. mRNA-seq of HL-1 cardiomyocytes was performed by Novogene. Total RNA quality was assessed by Bioanalyzer 2100, and poly(A)+ mRNA was enriched using oligo(dT) beads. Libraries were prepared using the NEBNext Ultra II RNA Library Prep Kit following the manufacturer's protocol and sequenced on an Illumina NovaSeq 6000 platform with paired-end 150 bp reads. For comparison of the separate groups, raw read counts were first normalized to correct for differences in sequencing depth and RNA composition. Differential expression analysis was then performed using DESeq2[51]. Dispersion estimates and fold changes were calculated for each gene, and significance was assessed through Wald or likelihood ratio tests depending on the model. Multiple testing correction was applied using the Benjamini–Hochberg procedure to control the false discovery rate (FDR). Genes with | $\log_2$(FoldChange)| >0.58496 and adjusted $p$ <0.01 were considered significantly differentially expressed.

## Western blotting
Protein lysates were collected from ~20 mg of powdered mouse hearts by mechanical homogenization in RIPA buffer supplemented with protease and deacetylase inhibitors. Homogenate was spun down at 4 °C for 10 min to pellet unbroken tissue, and the supernatant taken for further processing. Protein concentration was determined by Pierce BCA Kit, following recommended guidelines except with the addition of a prior dilution of aliquots by 1 part lysate to 4 parts RIPA buffer. Protein amount and concentration was then standardized in a new tube by aliquot of protein lysate and RIPA buffer, then prepared for denaturation with 4× Laemmli buffer and 10 min of boiling at 95 °C on heat block. 50 μg of protein was then loaded per well into a 10% acrylamide gel. Samples were allowed to linearize for 10 min at 70 V, before resolving at 120 V until the 15 kDa marker of the ladder (BioRad) ran to the bottom of the cassette. The gel was then removed and transferred to PVDF membrane via semi-dry transfer for 7 min (TurboBlot). The membrane was then assessed for total protein amount by Ponceau S staining for 1 min and imaging under the Chemidoc (BioRad). After rinsing with 1% NaOH to remove red stain and 10 min in TBST to stop reaction, the membrane was blocked for 1 h in 5% BSA in TBST, then incubated overnight in primary anti-Acetyllysine (SantaCruz, sc32268) antibody at 1/200 in 5% BSA-TBST. The membrane was washed 3× for 10 min each in TBST, then incubated for 2 h in secondary anti-mouse antibody at 1/10,000 in 5% BSA-TBST. Final revelation was performed for 7 s under the Chemidoc chemiluminescence imaging program after 30 s incubation in ECL substrate.

## Statistics and reproducibility
Statistical analyses were performed using Prism (GraphPad Software). All data are reported as mean ± s.e.m of distinct biological (individual mice in mouse experiments, separately cultured wells in cell experiments) replicates except for individual data points (representing separate wells of cultured primary cells from 1 mouse/condition/experiment run) in Fig. 1F. A 2-tailed unpaired Student's $t$ test was used when comparing two conditions (qRT-PCR and NAD$^+$ measurements). Comparisons of more than two conditions were done with One- or Two-way ANOVA followed by Tukey's multiple comparisons test when groups had comparable standard deviations; Brown–Forsythe ANOVA followed by Dunnett's T3 multiple comparisons test was used when standard deviations could not be assumed equal. Statistical analyses to detect circadian oscillations in PER2::luc levels (LV200) were performed by COSINOR[53]. For differential gene expression in the RNA-seq data, $p$ values were calculated via DESeq2[51].

## Reporting summary
Further information on research design is available in the Nature Portfolio Reporting Summary linked to this article.

## Data availability
The numerical source data for the graphs in this manuscript can be found in the Supplementary Data 01 file. The mRNA-seq raw and processed data has been deposited in the NCBI Gene Expression Omnibus (GEO) under accession number GSE240483, linked to BioProject PRJNA1002243. Further data can be made available upon reasonable request. Public domain clip art vectors used for Figs. 1A and 2D available at https://www.clker.com/clipart-white-mouse-black-ears.html.

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

## Acknowledgements

We kindly thank M.A. Lazar for providing us with some mouse material and K. Mattonet for help with the generation of a macro for image analysis. The authors thank Elysium Health for providing NR at no cost and J. Baur for the advice on the NR experiments. This work was funded by the Deutsches Zentrum für Herz-Kreislauf-Forschung e.V. (DZHK) (K.C., Y.N.M., P.D.) and the Cardiopulmonary Institute (CPI).

## Author contributions

P.D. conceived and designed the overall study. P.D., B.J.C., K.C., and S.G. contributed to next-generation sequencing experiments and bioinformatic analyses. Y.N.M. and B.J.C. performed NAD+ measurements. M.L. performed bioluminescence microscopy experiments. P.D. and B.J.C. performed gene expression measurements, cell culture experiments, animal caretaking, and lumicycle experiments. P.D., B.J.C. and M.W.V. wrote and edited the paper.

## Funding

## Competing interests

The authors declare no competing interests.
