## [Transparent Peer Review file · Communications Biology]

NAD⁺ controls circadian rhythmicity during cardiac aging

Corresponding Author: Dr Pieterjan Dierickx

Version 0:

Reviewer comments:

Reviewer #1

(Remarks to the Author)

Carpenter et al report relevant and important findings that bring together the investigation of circadian control of cardiac biology in the context of the ageing heart. The models are well-described and studies and statistical analyses are rigorously executed. The long-term supplementation study is particularly impressive and demonstrates the reversibility of several age-related disruptions in cardiac circadian rhythm.

Comments:

1. I understand that tissue is quite precious from the aged samples however it could be interesting to see pan-acetylation blots or the young versus aged versus aged+NR hearts to determine if any global changes could be detected (given the role of NAD⁺ in SIRT1 mediated deacetylation and its link to ageing). This could also be done in the Per2:Luc mice with and without FK866 for example.
2. I would like to see more discussion of the pathways rescued, or not, by NR in the aged heart and what the implications might be.
3. The studies in the isolated cardiomyocytes are very compelling. However, there should be discussion including the recent work by Basse et al (PMID: 36996103) that indicates the role of Nampt and NAD⁺ biosynthesis is highly tissue specific. It would seem the authors could add circadian control of the heart to this model.

Reviewer #2

(Remarks to the Author)

The paper by Bryce et al. describes the impact of NAD⁺ levels modulation on circadian gene expression rhythmicity which they show as globally affected during cardiac aging. The authors used nicotinamide riboside treatment, a precursor of NAD⁺, as an intervention to restore rhythmicity. The main mechanism proposed by the authors is an effect of NR on PER2 protein level rhythmicity. This is an interesting study, which, however, suffers from gaps in the mechanistic explanations of the observations made by the authors. For instance it remains unclear why some genes regain rhythmicity in NR-treated aged mice and other genes lose it. Moreover, several discrepancies between the sex and the age of the mice chosen for different the type of experiments weaken the significance of the conclusions that may be drawn from that study. In the following I raise some major points that should be addressed to improve the paper.

1/ The experiments are performed on females only and the justification is briefly detailed in the reporting summary that is not available to the reader. It's an important matter and the preliminary data that the authors alluded to about a difference between sexes in the response to NR should be explicated. Moreover the transcriptomic data are obtained from females while the PER2:luciferase assay were performed on males and at different ages. Considering the major differences in gene expression between males and females during aging (see for instance Yusifov et al. *Physiol Rep.* 2021. doi: 10.14814/phy2.14940), this is a major issue and the PER2:luciferase experiments should be performed using female cardiomyocytes.

2/ The Sirt1-dependant deacetylation of PER2 evoked in line 102 should be exemplified and confirmed in the present study in sex and age-matched animals relative to the rest of the study. This seems especially important as the transcriptomic analysis did not reveal any enrichment in Sirtuin pathway or more generally NAD signaling gene which may seem curious considering that the present study suggest it's a major mechanism in the loss of diurnally expressed genes in aged mice.

3/ NAD⁺ measurements are based on perchloric acid extraction that may not allow extraction of the whole NAD pool,

including NADH that is sensitive to acid pH. Therefore it's impossible to know if these data reflect a drop in global NAD pools or partially a shift from oxidized to reduced form of NAD. The authors should show the levels at ZT10 in addition to ZT22 to better understand the impact of NAD levels on these mechanisms. Same holds true for ANP expression data. The number of samples is too limited and the legend (line 179) indicates 4-5 whereas it's rather 3-4 only for 2A.

4/ Why the figure 2C are expressed as relative NAD⁺ levels whereas the panel 2A is in nmol/mg? Here again the number of sample is low and actually the pvalue is > 0.05.

5/ The effect of NR on PER:Luciferase signal in cultured cardiomyocytes is performed in neonate cardiomyocytes and not adults as in previous figure. It does not fit with the message of the paper on the importance of aging on circadian gene oscillation in the heart. Moreover, a brief analysis of the supplementary data provided by the authors do not show any change in Nampt gene expression in young and old mice and at the 2 ZT time point. On the other hand, the expression of Nmrk2 that metabolizes NR seems to be lower in aged mice. Hence the rationale of Nampt inhibition in cultured cardiomyocytes to study the effect of NR does not fit with the in vivo data. The authors should discuss these points in their study.

Minor point

1/ The introduction is not providing enough on cardiac ageing specifically and what is known about NAD signaling and biosynthetic pathways (NAMPT, NMRK...) in this organ.

2/ Line 67, it would be clearer to indicate "in the present study" to make it clear that this is now the results of the study that are summarized.

3/ The choice of the 2 time points used throughout the study is justified by data from Mia et al 2023 shown in supplementary figure 1. However, those data were obtained on males and not females that were used in the present study and the age is different. Can the authors provide better justification for the choice of these time points?

4/ The authors do not precise the C57Bl6 substrain. Are they Bl6j or Bl6n? This is important as Nickel et al (Cell Metab. 2015; doi: 10.1016/j.cmet.2015.07.008) have shown that the 2 strains that differ notably by a mutation in Nnt gene, which is important in NAD metabolism, behave differently in response to cardiac stress.

5/ Fig 1B show of the numbers of "diurnally" expressed genes. It's not very clear for the reader if this numbers represent the sum of up and downregulated genes during day time only, or both day and night. Numbers could be added to figure 1C to better understand the size of the clusters. Moreover, it seems that only the FDR <0.05 was used as a cut-off. Could the authors indicate how many genes are modulated relative of fold change cut-off to provide a better understanding of the level of changes.

6/ Figure 2I, Student test does not seem to be appropriate considering that there are 6 groups with different age and treatments. A one-way anova would be more appropriate or eventually a 2 way anova is focusing on the old mice.

Reviewer #3

(Remarks to the Author)

In this manuscript, Carpenter and colleagues report on the rescue of clock dampening in heart during later adulthood in mice (around 1 year of age) by rescuing NAD⁺ abundance. The causative non-genetic mechanisms regarding overall NAD⁺ abundance (NR supplementation vs FK866 inhibition), as well as the unbiased mRNA oscillation analysis through RNA-seq make this report a compelling addition to the field. Parallel analyses with PER2:Luciferase+ cardiomyocytes provide an additional strength. The manuscript is very interesting, and I would like to ask the Authors to address some additional points to further substantiate their findings:

Major points:

- Were ADP ribosylation and PARP activity/expression changed by age/treatments in hearts or cardiomyocytes?
- Did age/treatments affect the NAD⁺/NADH ratio in hearts or cardiomyocytes? Related to that, were there changes in redox balance and energy production?
- Were molecular markers of PER2 hyperacetylation changed? E.g. Per2 AcK, nuclear coIP with CRY, lowered BMAL1 activity on canonical sites, ...etc?
- heart weight to body weight data are encouraging, yet quite uninformative. Were those reflective of aging-related myocardial hypertrophy? Minimal echocardiography or histological data will be sufficient to further compound this point. I think this is critical because it speaks to the extent of "cardiac aging" actually displayed by the mice used in this manuscript.
- (related to Suppl Fig 1) a chart related to diurnal clock gene expression difference from other datasets (although from the same group) seems unnecessary considering the RNAseq datasets generated for this very manuscript. Charts from the RNAseq data collected for this manuscript will better illustrate the level of derangement/rescue of the clock gene oscillations in the actual hearts used here.

Minor points:

- 12-15 months is approximately mid-life for WT mice in lab captivity. Therefore ~1yr-old mice are not generally considered a strong model of aging or – to be more precise – geriatric conditions. That said, the data and the interpretation logic are valid

and unaffected by this consideration. However, the overall significance for heart aging could be impacted. Therefore, I would recommend the Authors to address this in the manuscript by 1) adding some rationale to the age used in the Results section, and 2) add some speculative considerations regarding geriatric age settings in the Discussion.

Version 1:

Reviewer comments:

Reviewer #1

(Remarks to the Author)

I would like to commend the authors are thoroughly addressing my comments. I believe these points have strengthened an already exciting manuscript and I have no further concerns.

Reviewer #2

(Remarks to the Author)

The authors have provided a clearly improved manuscript and extended their finding by adding additional experiments based on the request of the reviewers. I truly appreciate their efforts and the paper is bringing a significant amount of interesting new information. However, there are still a few points that would deserve attention to my point of view.

Major points

1/ I appreciate the effort of the authors to repeat their experiments in female cardiomyocytes rather than male cardiomyocytes to better fit with their transcriptomic analyses and previous studies focusing on females. Previous data on male cardiomyocytes could still be shown in supplementary data as they support the model.

2/ The authors are honestly showing in supplement Figure 1B and C, that a second run of experience yielded a less robust oscillation in adult female cardiomyocytes. This reviewer understands these experiments that are performed for 36 h on adult mouse cardiomyocytes are difficult and agree that altogether the results essentially corroborate their *in vivo* data. Never the less, it would be more correct to show in the main figure a pool analysis of the 2 experiences, even if *p*val is above 0.05, the trend should be there. Or maybe, if the Editor concurs, it could be possible to pool results from males and females for more statistical power and then show details in supplement to highlight individual experiences in supplementary figures, showing that the technique is subjected to batch variation. Ideally a 3rd run of experiment on the female cardiomyocytes would be good. Eventually I suggest adding BDM or blebbistatin to the cell culture medium to minimize cell rounding over time if it does not affect too much the response of course.

3/ I also appreciate the effort put by the authors to analyze NMRK2 protein level in the heart and it could be interesting to show it. However, mouse NMRK2 predicted size is 22,375 kDa (Q9D7C9 NRK2_MOUSE in UNIPROT database), hence it is very close to light chain IgG at 25 kDa that can be present in mouse tissues. As the antibody that the authors used is a mouse monoclonal requiring the use of secondary anti-mouse IgG, there is always a risk to reveal IgG. The band shown in gel in the FigC of rebuttal is unfortunately too fuzzy to convincingly demonstrate it is NMRK2. The authors could use gels dedicated to small size proteins to get sharper bands and assess presence or absence of IgG in their protein extract using 2ndary antibody only. However, if this is too difficult because samples are missing, I don't consider this as a key point for the article.

4/ This reviewer agrees that the use of neonate cardiomyocytes that can be cultured over a longer period to study how NAD⁺ modulation affects PER2 rhythmicity is a good idea. And actually, the results are nice but it would seem important to show the level of NAD⁺ reduction.

5/ The rationale for switching to HL1 cells after having established the neonate rat cardiomyocytes model is not clear (and not justified in the text). Ex527 works fine in those cells too and HL1 cells not a very good model of cardiomyocytes. And beyond this, the transcriptomic data do not show the impact of FK866 compared to controls, which seems strange, as well as the lack of pathway enrichment analysis on these data

Minor points:

1/ Line 84 and 137: probably reference 15 rather than 19

2/ For a better global understanding of the data, a short sentence could remind the rationale for performing detrended fluctuation analysis at least in material and methods.

3/ Since the NADH data are available for the cell experiments, it's always better to show it.

Reviewer #3

(Remarks to the Author)

The Authors addressed all points from me and other Reviewers with additional analyses and experiments when possible.

Version 2:

Reviewer comments:

Reviewer #2

(Remarks to the Author)

The authors addressed all points carefully and to the best of their ability. The paper is now very clear and convincing. Congratulation for this excellent study.

Response to Editor's and Reviewer's comments

Title: NAD⁺ controls circadian rhythmicity during cardiac aging

Authors: Bryce J. Carpenter, Margaux Lecacheur, Yannick N. Mangold, Kai Cui, Stefan Günther, Pieterjan Dierickx

Manuscript ID: COMMSBIO-23-3060-A

General comments to the editor and reviewers

The authors are grateful to the editor and the reviewers for their constructive review comments. The manuscript has been revised in response to these comments, as outlined in detail below.

The authors believe that the revised manuscript has improved significantly as an immediate result of the review process. Importantly, we are confident that the revised manuscript provides novel insight in multiple ways, such as revealing that: 1) female cardiomyocytes reprogram their circadian clock upon aging and treatment with nicotinamide riboside; 2) SIRT1 plays a partial role in circadian reprogramming upon NR; and 3) naturally occurring myocardial hypertrophy upon long-term NR treatment can be reversed.

General comments for Reviewer 1

"Carpenter et al report relevant and important findings that bring together the investigation of circadian control of cardiac biology in the context of the ageing heart. The models are well-described and studies and statistical analyses are rigorously executed. The long-term supplementation study is particularly impressive and demonstrates the reversibility of several age-related disruptions in cardiac circadian rhythm."

The authors would like to thank the reviewer for the complimentary comments.

Specific comments for Reviewer 1

1. *I understand that tissue is quite precious from the aged samples however it could be interesting to see pan-acetylation blots or the young versus aged versus aged+NR hearts to determine if any global changes could be detected (given the role of NAD⁺ in SIRT1 mediated deacetylation and its link to ageing). This could also be done in the Per2:Luc mice with and without FK866 for example.*

We would like to thank the Reviewer for this suggestion. On this recommendation, we did perform the requested pan-acetylation blot in the originally treated samples, which is now included in the new figure 5E. An increase in acetylation by age was easily captured, but not any significant restoration from NR treatment. In addition, we attempted some blots with our limited protein for known targets or suspected mechanisms, such as the level of PER2 or the acetylation state of particular marks, but failed to concretely validate targets due to the low quality of the tested commercial antibodies. See comments in response to reviewer 2, major comment 2 for images and further context of the specific strategies taken.

2. *I would like to see more discussion of the pathways rescued, or not, by NR in the aged heart and what the implications might be.*

While these pathways still require more mechanistic probing to draw full conclusions, we did add greater emphasis in the results to the correlation between FoxO signalling genes that are returning to a rhythmic state with NR treatment, including the potential for SIRT1 interaction with this pathway in the heart. Notably, the factors suggested by LISA to enact these transcriptional changes include cardiac development factors, like TBX3 and JARID2. While the development of rhythmicity in this instance cannot yet be correlated as either detriment or benefit to cardiac function, it should also be recognized that our top transcription factors for novel oscillators are known mediators of cardiac hypertrophy (MEF2A and SUZ12). These insights, including background references, have now been included in the final paragraphs of the Results section.

3. *The studies in the isolated cardiomyocytes are very compelling. However, there should be discussion including the recent work by Basse et al (PMID: 36996103) that indicates the role of Nampt and NAD⁺ biosynthesis is highly tissue specific. It would seem the authors could add circadian control of the heart to this model.*

This point is taken in well regard, and we have briefly expanded in the conclusion on the different known reactions to NAD⁺ modulation in liver, brown adipose tissue, and skeletal muscle, with reference to the paper by Basse et al. Of note but not yet explored fully, these factors may also be interrelated; as aging, NAD⁺ pathways, and circadian regulation vary by tissue, the interactions of these systems may contribute to unique transcriptomic and physiological states by cell type or organ, as elaborated on in the discussion of both this paper and the work by Basse et al.

General comments for Reviewer 2

“The paper by Bryce *et al.* describes the impact of NAD⁺ levels modulation on circadian gene expression rhythmicity which they show as globally affected during cardiac aging. The authors used nicotinamide riboside treatment, a precursor of NAD⁺, as an intervention to restore rhythmicity. The main mechanism proposed by the authors is an effect of NR on PER2 protein level rhythmicity. This is an interesting study, which, however, suffers from gaps in the mechanistic explanations of the observations made by the authors. For instance it remains unclear why some genes regain rhythmicity in NR-treated aged mice and other genes lose it. Moreover, several discrepancies between the sex and the age of the mice chosen for different the type of experiments weaken the significance of the conclusions that may be drawn from that study. In the following I raise some major points that should be addressed to improve the paper.”

The authors would like to thank the reviewer for the complimentary comments.

Specific comments for Reviewer 2

1/ The experiments are performed on females only and the justification is briefly detailed in the reporting summary that is not available to the reader. It's an important matter and the preliminary data that the authors alluded to about a difference between sexes in the response to NR should be explicated. Moreover the transcriptomic data are obtained from females while the PER2:luciferase assay were performed on males and at different ages. Considering the major differences in gene expression between males and females during aging (see for instance Yusifov *et al. Physiol Rep.* 2021. doi: 10.14814/phy2.14940), this is a major issue and the PER2:luciferase experiments should be performed using female cardiomyocytes.

We thank the reviewer for the additional enlightening reference; we have included more explicit discussion of our decision to use female mice, including our previous work showing the efficacy of NR treatment on female mice in a disrupted clock model (PMID: 37389009). Additionally, we have repeated the isolation of PER2::Luc reporter cardiomyocytes from aged and young female animals to better match our transcriptomics data, and included two independent performances of this experiment in the manuscript and supplementary data (New Figure 1F, and New Supplemental Figure 1B). Without the current means to expand the study to include direct and appropriate comparison between male and female mice across both ages in all experiments, we have elected to remove the male data of aged vs young hearts. In the meantime, we have shifted the focus of the experiments to accentuate and clarify that the results presented should only be taken in the context of aging female mice, with no specific claim to male mice in this context.

2/ The Sirt1-dependant deacetylation of PER2 evoked in line 102 should be exemplified and confirmed in the present study in sex and age-matched animals relative to the rest of the study. This seems especially important as the transcriptomic analysis did not reveal any enrichment in Sirtuin pathway or more generally NAD signaling gene which may seem curious considering that the present study suggest it's a major mechanism in the loss of diurnally expressed genes in aged mice.

While the acetylation status of PER2 is a promising possibility given the strong evidence presented in Asher *et al* (PMID: 18662546) and Levine *et al* (PMID: 32369735) for the liver, we did not mean to indicate that this is necessarily the case in the heart or in this study. SIRT1 has been shown to interact with the clock through other mechanisms, and the myriad enzymes making use of NAD⁺ may also be capable of feeding into the clock and/or cardiac hypertrophy. As such, we have added more discussion of other NAD⁺ consumers as well as previously known functions of SIRT1 in association with the clock. However, in pursuing this interesting point and from comments from other reviewers, we also tried to assess the dynamics of PER2 acetylation status via two methods; immunoprecipitation of acetylated residues and blotting for PER2, as well as IP-MS of total acetylated residues and hoping to find PER2 peptides. Unfortunately, the commercially available antibodies for PER2 failed to confidently validate visualization of PER2, expected due to low abundance and difficult size (~140kDa), in addition to known poor quality of the antibodies themselves. Likewise, we are working with our resident mass spectroscopy facility to optimize an acetylomics protocol on heart powder, but we have only middling success in developing our own sufficient in-house IP protocol, while the facility has struggled to achieve comprehensive acetylomics using the commercially available Cell Signaling Kit on other heart projects. Despite these challenges in capturing PER2 specifically, we performed new experiments to further accentuate the role of SIRT1 in the NR transcriptional response by blocking its activity with small molecule Ex527, and discovered a subset of genes that only respond to NR from a low NAD⁺ state if SIRT1 is active. This new data has been incorporated into the main body as part of New Figure 5.

Figure A/B in response to Reviewer 2, comment 2. PER2 Western blot, samples prepared as described in manuscript, with only antibody and exposure time changed. Samples were taken from new circadian collections performed on female WT mice at the ages corresponding to the manuscript. Each figure denotes a separate blot. A) MBL Life Science product PM083 at 1/200 dilution, exposure time of 2min. B) top, Santa Cruz Biotechnologies product sc-377290 at 1/200 dilution, exposed for 1min; bottom, performed on the same blot as top after stripping/reprobing, AbClonal product A5107 at 1/1000 dilution, exposure of 30min. Blue arrow indicates approximate location of anticipated size for a PER2 band, with highest protein expression expected at ZT18.

Figure C in response to Reviewer 2, comment 2. Immunoprecipitation of acetylated proteins in WT heart tissue. Protein lysates including 700 μ g protein, prepared as described in the manuscript, were incubated with 10 μ g anti-acetyllysine antibody (sc-32268) at 4C for 48hrs, then with 50 μ L magnetic ProteinG beads (New England Biotechnologies, S1430S) for 2hrs. Supernatant was removed and the beads washed 3x in lysis buffer, then boiled directly in Laemmli buffer for elution. Western blot proceeded as described in manuscript, on the same anti-

acetyllysine antibody. Input = 5% Input samples (protein lysate prior to antibody incubation), IP = immunoprecipitated samples, Sup = supernatant from bead incubation after antibody capture.

3/ NAD⁺ measurements are based on perchloric acid extraction that may not allow extraction of the whole NAD pool, including NADH that is sensitive to acid pH. Therefore it's impossible to know if these data reflect a drop in global NAD pools or partially a shift from oxidized to reduced form of NAD. The authors should show the levels at ZT10 in addition to ZT22 to better understand the impact of NAD levels on these mechanisms. Same holds true for ANP expression data. The number of samples is too limited and the legend (line 179) indicates 4-5 whereas it's rather 3-4 only for 2A.

Indeed, the complex steps of the NAD synthesis and consumption pathways include several metabolites and changes that would not be reflected by solely checking NAD⁺ levels, including redox status. Because our previous protocol, as stated, relies on acidic extraction and thus requires separate aliquots of material for analyzing even just the addition of NADH, we have worked in approaches to reform our measurements. Firstly, as now presented in the paper, we have developed a protocol adjusted from the Promega NAD⁺/NADH Glo Assay, designed around use on various cell lines after plating, to work on primary cells, and can now feasibly measure both NADH and NAD⁺ from a shared cell sample of relatively little material per biological replicate. However, to follow through with protein analysis and other recommendations from the comments for revision, there is not enough material from the original cohort to perform the adapted assay, and generation of a new cohort is in the process of gaining approval. Since new performances of all assays across every condition is not feasible, we have elected to retain the original ZT22 NAD⁺ measurement on heart tissue, and have included the changed protocol in the measurement of our HL1 cell line with only NAD⁺ measurements to correspond to the first *in vivo* cohort. However, with the data available from the RNA-Seq, we have now forgone the original qPCR and expanded the *Anp* expression data in the main figure to include both measured timepoints, though the inclusion of *Anp* and *Bnp* was primarily to correspond to global marker of physiological cardiac health, rather than of rhythmic maintenance.

Figure A/B in response to reviewer 2, comment 3. NAD⁺ and NADH levels in HL1 cells used for RNA-Seq analysis, reconfigured from the new data of manuscript Figure 5. Assay performed per adapted instructions from supplier as stated in main methods.

4/ "Why the figure 2C are expressed as relative NAD⁺ levels whereas the panel 2A is in nmol/mg? Here again the number of sample is low and actually the pvalue is > 0.05."

We believe the relative values make a cleaner presentation of the fold changes between conditions, especially in the possibility of standard curve or plate variability between runs, and so have updated both figures to relative values. We have ensured that the *p*-value is clearly presented as to not mislead, and included in the new supplemental (**SuppFig2**) evidence that the NR treatment was extremely effective at increasing NAD⁺ levels in the liver; relatively less efficacy is likely due to first-pass metabolism, but downstream metabolites, turnover changes, or cross-organ uptake would not be reflected in solely NAD⁺ levels.

5/ "The effect of NR on *PER:luciferase* signal in cultured cardiomyocytes is performed in neonate cardiomyocytes and not adults as in previous figure. It does not fit with the message of the paper on the importance of aging on circadian gene oscillation in the heart. Moreover, a brief analysis of the supplementary data provided by the authors do not show any change in *Nampt* gene expression in young and old mice and at the 2 ZT time point. On the other

hand, the expression of *Nmrk2* that metabolizes NR seems to be lower in aged mice. Hence the rationale of *Nampt* inhibition in cultured cardiomyocytes to study the effect of NR does not fit with the *in vivo* data. The authors should discuss these points in their study.”

We thank the reviewer for pointing this out. We have further clarified the distinct conclusions these experiments were intended to assess in the summary; the experiments performed with neonatal cardiomyocytes and FK866 were not intended as direct models of aging, but to circumvent the confounding difficulties of long-term treatment and culture of adult cardiomyocytes. Instead, FK866 was used to induce a state of lower NAD⁺, as we have demonstrated to be the case over aging (regardless of which changes to biosynthesis pathways determine such conditions). Assuredly, neonatal cardiomyocytes are quite distinct from either level of maturation used in the paper, but are generally more susceptible to sustained culture and less plastic than adult cardiomyocytes. We attempted to perform the modulations of NAD⁺ via NR and FK866 *in vitro* with cardiomyocytes isolated from appropriately aged mice, but adult cardiomyocytes lose their native morphology after 3 days, becoming rounded and weakening adherence to the plate. In this cell type and current *in vitro* conditions, cells did not respond within this time frame to the small molecules in a detectable and quantifiable manner, limiting our data collection to the more tolerable neonatal cardiomyocytes.

While *Nampt* and *Nmrk2* expression and activity are highlighted pathways for the generation of NAD⁺, it is well-documented that the loss of NAD⁺ over age could be due to differences in synthesis, redox status, consumption, or other avenues, and that the responsible enzymes have dynamic levels of function after their transcription. Thus, the picture taken of the maintenance or alteration of NAD pathways by only gene expression is incomplete, but we have now included a highlight of those pathways in the manuscript as a first step in demystifying the cause of lower NAD⁺ in age. We did also evaluate NAMPT protein expression in our original mouse samples, and validated that there is no loss of NAMPT protein between them at this level of aging. Other studies that have reported the loss of NAMPT (or commiserate gain of NMRK2) were performed in more severe aging models, and thus may indicate a different and/or complementary source of aging-related NAD⁺ change. We also sought to examine NMRK2 protein expression, and despite the product page of our antibody displaying a size near 35 kDa, the only band that appeared (close to the actual predicted size of 26kDa) was increased in older mice. As conflicted these results may be, the fact remains that the cardiac NAD⁺ level is still lower at 1 year of aging, and reflecting that condition was the aim of the FK866 usage in cell experiments. To clarify the point we wish to make: the heart loses some degree of NAD⁺ levels, expresses minor hypertrophy, and substantially shifts its rhythmic output even by ~1 year of age; supplementation of NR prevents the hypertrophy and puts a fair amount of rhythmic output back to a younger state; neonatal cardiomyocytes show that lower NAD⁺ levels and restoration with NR is able to maintain normal rhythmicity, as defined by transcription for output genes (partially via SIRT1 as performed in HL1 cells) and protein tracking in the core clock.

Figure A in response to Reviewer 2, comment 5. Decrease in viability and structured morphology of isolated adult cardiomyocytes after three days, as assessed by our size calculation methodology. As such, any adult CM experiments were limited to three days of data collection at most.

Figure B in response to Reviewer 2, comment 5. Western blot for NAMPT protein expression in the Young v Old v NR-Treated cohort from ZT10-collected samples. NAMPT presents with a 2-3 banding pattern at 55kDa. Protein lysate and western blot performed as described in main manuscript, with primary antibody changed. Top, anti-NAMPT antibody (Bethyl product A300-372A) at 1/1000 in 5% Milk-TBST, exposure time of 30sec. Bottom, anti-Vinculin conjugated to HRP (Cell Signaling Technology product E1E9V) 1/1000 in 5% Milk-TBST with no secondary antibody incubation, exposure time of 8sec.

Figure C in response to Reviewer 2, comment 5. Western blot for circadian NMRK2 protein expression in the newly collected Young v Old cohort. NMRK2 presents at an estimated running weight of 26-36kDa. Protein lysate and western blot performed as described in main manuscript, with primary antibody changed. Top, anti-ITGB1BP3 antibody (Origene product CF813057) at 1/1000 in 5% BSA-TBST, exposure time of 20sec. Bottom, anti-Vinculin conjugated to HRP (Cell Signaling Technology product E1E9V) 1/1000 in 5% Milk-TBST with no secondary antibody incubation, exposure time of 6sec.

Minor point

1/ *“The introduction is not providing enough on cardiac ageing specifically and what is known about NAD signaling and biosynthetic pathways (NAMPT, NMRK...) in this organ.”*

We acknowledge the reviewer’s concern. Both topics are of course critical to the proper contextualization and understanding of our manuscript. As such, we have extensively added to the introductory paragraphs, giving more initial focus to the process of cardiac ageing, particularly in regards to metabolism. We believe this creates a more natural flow into the description of the many functions of NAD⁺ prior to connecting to the more niche topic of circadian rhythmicity on these subjects.

2/ *“Line 67, it would be clearer to indicate “in the present study” to make it clear that this is now the results of the study that are summarized.”*

We thank the reviewer for highlighting this. The text has been changed from “Here” to say “In the present study” for clarification.

3/ "The choice of the 2 time points used throughout the study is justified by data from Mia et al 2023 shown in supplementary figure 1. However, those data were obtained on males and not females that were used in the present study and the age is different. Can the authors provide better justification for the choice of these time points?"

New supplemental figure 1 has been updated to now show qRT-PCR data generated in-house on 12-week-old female mice, therefore matching more to the young (2-3months) experimental groups presented elsewhere in the paper. These particular timepoints are broadly used throughout the literature, as the protein and RNA peak/trough expression of the primary clock pathways are very robust in unperturbed conditions.

4/ "The authors do not precise the C57Bl6 substrain. Are they Bl6j or Bl6n? This is important as Nickel et al (Cell Metab. 2015; doi: 10.1016/j.cmet.2015.07.008) have shown that the 2 strains that differ notably by a mutation in Nnt gene, which is important in NAD metabolism, behave differently in response to cardiac stress."

The selection of animals as described in the "Methods" section now specifies that the mice are a diffused mix of Bl6J/N, representing a heterogenous population more broadly applicable to interpretation of previous NR trials. However, we do not know of any previous publications directly comparing the effects of C57Bl6J vs N in response to NR treatment/the *Nmrk2* pathway specifically. Notably, experiments identifying phenotypical differences between the homozygous Bl6J and Bl6N strains rely on much more severe and pathological cardiac conditions such as pressure overload as in the reference above.

5/ "Fig 1B show of the numbers of "diurnally" expressed genes. It's not very clear for the reader if this numbers represent the sum of up and downregulated genes during day time only, or both day and night. Numbers could be added to figure 1C to better understand the size of the clusters. Moreover, it seems that only the FDR <0.05 was used as a cut-off. Could the authors indicate how many genes are modulated relative of fold change cut-off to provide a better understanding of the level of changes."

The usage of "diurnally" in our instance was to indicate genes with significantly different expression between day and night; for clarification we have updated the title of **New Fig. 1B**. We have also added counts to the left side of **New Fig. 1C** to now match those in the headers of **Fig1D**. The methods section was updated to clearly state there was no applied FC cutoff after the adjusted *p*-value calculation. However, every gene represented has a $\log_2(\text{FC}) > 0,18$.

6/ "Figure 2I, Student test does not seem to be appropriate considering that there are 6 groups with different age and treatments. A one-way anova would be more appropriate or eventually a 2 way anova is focusing on the old mice."

There is no additional statistical analysis done in **Fig. 2I** (now **Fig. 3C**); these 6 groups are not being directly compared by numbers, as these graphs are instead to show counts for representative genes of each previously identified class (rhythmic in all, rhythmic only after NR treatment, etc.). As such, the only statistical analysis performed on this subfigure is the initial calling of significance by ZT10 vs ZT22 counts (DESeq2) within each age/treatment group. The stars in the figure and thus the Student t-test pertain only to **Figure 2**, while the remaining claims of important differences are demarcated by the rhythmicity sign, with assignment on the left side of the graph determined by whether they pass cutoff values within their respective groups (e.g., FDR < 0.05 in young, but not in old, is called as "young only" diurnal).

General comments for Reviewer 3

"In this manuscript, Carpenter and colleagues report on the rescue of clock dampening in heart during later adulthood in mice (around 1 year of age) by rescuing NAD⁺ abundance. The causative non-genetic mechanisms regarding overall NAD⁺ abundance (NR supplementation vs FK866 inhibition), as well as the unbiased mRNA oscillation analysis through RNA-seq make this report a compelling addition to the field. Parallel analyses with *PER2:luciferase+* cardiomyocytes provide an additional strength. The manuscript is very interesting, and I would like to ask the Authors to address some additional points to further substantiate their findings:"

The authors would like to thank the reviewer for the complimentary comments.

Specific comments for Reviewer 3

Major points:

“- Were ADP ribosylation and PARP activity/expression changed by age/treatments in hearts or cardiomyocytes?”

This question is appreciated, given the role of PARPs in consuming NAD⁺ as a possible method for the loss with age. As we had narrowed our search and claims in particular to changes in rhythmicity/diurnal changes to expression, it was not a particular focus in our first search. Despite several PARPs being found throughout all the replicates and groups, there seems to be very little in the way of rhythmicity or response to this age or treatment strategy. This is an interesting contrast to the liver, where PARP1 has been shown to strongly interact with entrainment due to feeding patterns, and may highlight a substantial difference for the circadian regulation of NAD⁺ between the two tissues studied. Of course, protein or activity levels may not correspond directly to mRNA expression, but with limitations to the materials and relative abundance of literature on sirtuins in the cardiac context, we cannot follow-up on this discrepancy within the scope of this paper.

Figure A in response to Reviewer 3, comment 1. Heatmap of Z-normalized RNA-Seq expression data for all *Parp* counts found.

Figure B in response to Reviewer 3, comment 1. Graph of RNA-Seq counts for *Parps* with differential expression between any timepoints or ages. *p < 0.05, **p < 0.01, ***p < 0.005, ****p < 0.001 by Two-way ANOVA with Tukey’s multiple comparisons test.

“- Did age/treatments affect the NAD⁺/NADH ratio in hearts or cardiomyocytes? Related to that, were there changes in redox balance and energy production?”

Unfortunately, our original protocol for the assessment of NAD⁺ levels precluded using the same material for NADH measurement, and we could not spare the material immediately to rectify that in the original cohort. We have since adjusted a commercial protocol for plated cells to work on NAD⁺ and NADH within the same lysis from primary cells and tissues, but the remaining material was decidedly used to try to address protein assessments in prioritization of solidifying evidence towards a mechanism, as it was a point brought by all reviewers. For further context, please refer to comment 3 of Reviewer 2. While it would have been great foresight to test more thoroughly for mitochondrial changes in our treated group, including inspection of the mitochondria themselves, we have only so far been able to follow up with WT circadian cohorts at various ages without NR treatment; WB for oxidative phosphorylation markers show little difference over age or rhythmically. However, much in line with the comment on PARPs above, stronger assessment of the entire redox and metabolic status once we have successfully generated a new (live) treated cohort is at the top of the list.

Figure A in response to Reviewer 3, comment 2. Western blot for a collection of the oxidative phosphorylation proteins ATP5A (55kDa), SDHB (30kDa), UQCRC2 (48kDa), and NDUFB8 (20kDa). Protein lysate and western blot performed as described in main manuscript, with primary antibody changed to Total OXPHOS Rodent antibody cocktail (Abcam product ab110413) at 1/250 in 5% Milk-TBST. Exposure time of 9 seconds.

“- Were molecular markers of PER2 hyperacetylation changed? E.g. Per2 AcK, nuclear coIP with CRY, lowered BMAL1 activity on canonical sites, ...etc?”

While this set of questions was of great intrigue to us, we ultimately failed to sufficiently capture PER2 for any of these analyses. We tried first to capture PER2 by standard western blot, but due to well-known poor quality of the commercially available antibodies, could never confirm the presence of mouse PER2. We finally also tried to develop an acetylomics protocol with our mass spectrometry department, but the progression is very slow as our in-house IP for total acetylated peptides was insufficient and they are struggling to adjust the Cell Signaling Kit for use on heart powder, amongst their other projects at the institute. In addition to other findings throughout the revision process, we have largely stepped away from demonstrating the PER2 hypothesis, instead choosing to focus our efforts on further characterization of the role of sirtuins themselves in mediating the response to NR, rather than specifically through the PER2 axis at this time. For images of the trial blots, please refer to comment 2 from Reviewer 2.

“- heart weight to body weight data are encouraging, yet quite uninformative. Were those reflective of aging-related myocardial hypertrophy? Minimal echocardiography or histological data will be sufficient to further compound this point. I think this is critical because it speaks to the extent of “cardiac aging” actually displayed by the mice used in this manuscript.”

We thank the reviewer for this suggestion. Unfortunately, the necessity to repeat the treatment process is rather limiting for the evaluation of histology or gross physiological parameters in the scope of this paper. To address this, we further looked into known cardiac hypertrophy markers expressed in our dataset and have added their RNA-Seq counts below, with further emphasis on transcription factors brought up in response to Reviewer 1, comment 2. Additionally, we started working with cardiomyocyte cells lines (HL1) and using small molecules to focus on the low NAD⁺ environment and its reliance on SIRT1, to answer other mechanistic questions presented to us and now presented within the main figures. Despite the current inability to perform further evaluation without the mouse

treatment available, we still believe the significant depletion of rhythmic genes over our aging model indicates a prominent effect as a consequence of this degree of aging.

Figure A in response to Reviewer 3, comment 4. RNA-Seq counts for predicted transcription factors mediating hypertrophic and NR treatment response. In particular, MEF2A and SUZ12 are known hypertrophy factors, while TBX3 and JARID2 are both associated with cardiomyocyte growth and development. For clarity of view, only significant pairwise comparisons are displayed. * $p < 0.05$; ** $p < 0.01$; *** $p < 0.001$; assessed by 2-way ANOVA with Tukey's multiple comparisons adjustment.

Figure B in response to Reviewer 3, comment 4. RNA-Seq counts for classical hypertrophy markers not discussed elsewhere in the manuscript. For clarity of view, only significant pairwise comparisons are displayed. * $p < 0.05$; ** $p < 0.01$; *** $p < 0.001$; assessed by 2-way ANOVA with Tukey's multiple comparisons adjustment.

“- (related to Suppl Fig 1) a chart related to diurnal clock gene expression difference from other datasets (although from the same group) seems unnecessary considering the RNAseq datasets generated for this very manuscript. Charts from the RNAseq data collected for this manuscript will better illustrate the level of derangement/rescue of the clock gene oscillations in the actual hearts used here.”

The initial use of the graph from Mia et al. was to justify and provide an example for the particular choices of ZT10 and ZT22 for the RNA-Seq and experimental timepoints; we believed this point was made stronger by showing a higher time resolution of the expression pattern of clock components, since our RNA-Seq was restricted to only the two target timepoints. However, based also on minor comment 3 from Reviewer 2, we have replaced Supplemental

Fig. 1 with qRT-PCR data from young female mice, also generated in our own lab, to relieve any concerns about the relevance of the selected timepoints.

Minor points:

“- 12-15 months is approximately mid-life for WT mice in lab captivity. Therefore ~1yr-old mice are not generally considered a strong model of aging or – to be more precise – geriatric conditions. That said, the data and the interpretation logic are valid and unaffected by this consideration. However, the overall significance for heart aging could be impacted. Therefore, I would recommend the Authors to address this in the manuscript by 1) adding some rationale to the age used in the Results section, and 2) add some speculative considerations regarding geriatric age settings in the Discussion.”

We appreciate the nuance and insight of this response; undoubtedly, the phenotypic results could be expected to become more apparent or robust with further aging. However, at the time of the experiment, no other study into the effects of NR supplementation on the heart had utilized such a long treatment span. Given that we were already able to detect lower NAD⁺ levels by 15 months old, we thought this would prove a sufficient point for confirming the efficacy of the extended treatment strategy and provide initial direction to NAD-related cardiac aging patterns and pathways without unnecessary waiting. The anticipated consequence is, of course, that the pathological phenotype itself is less prominent than in advanced geriatric conditions, but provides a more graduated approach to the process of aging. Still, we are happy to posit more grounded speculation regarding this phenotype and the promising usage of NR, and have included such per this suggestion throughout the manuscript. Likewise, we present rebuttal data in response to Reviewer 2, comment 5 to further accentuate some of the protein changes in our 1-year cohort.

Response to Editor's and Reviewers' Comments

Title: NAD⁺ controls circadian rhythmicity during cardiac aging

Authors: Bryce J. Carpenter, Margaux Lecacheur, Yannick N. Mangold, Kai Cui, Stefan Günther, Pieterjan Dierickx

Manuscript ID: COMMSBIO-23-3060-A

General Comments to all

The authors are grateful for the continued time and attention of the editor and reviewers, particularly in the enthusiastic support for our initial work and first round of revisions. We have again taken this feedback in stride, and detail our responses to the received comments below.

General Comments for Reviewer 1

"I would like to commend the authors are thoroughly addressing my comments. I believe these points have strengthened an already exciting manuscript and i have no further concerns."

The authors are glad that our efforts were seen as valuable additions by the Reviewer, and appreciate the excitement for the work we have accomplished here.

General Comments for Reviewer 2

"The authors have provided a clearly improved manuscript and extended their finding by adding additional experiments based on the request of the reviewers. I truly appreciate their efforts and the paper is bringing a significant amount of interesting new information. However, there are still a few points that would deserve attention to my point of view."

The authors are very thankful to this Reviewer for continuously guiding significant steps forward in the strength and logic of our study, and hope that we have sufficiently addressed their constructive feedback below.

Specific Comments for Reviewer 2

Major

1/ I appreciate the effort of the authors to repeat their experiments in female cardiomyocytes rather than male cardiomyocytes to better fit with their transcriptomic analyses and previous studies focusing on females. Previous data on male cardiomyocytes could still be shown in supplementary data as they support the model.

The authors agree that aligning the sex of the mice between the RNA-seq and the LV200 experiments is a substantial and clear improvement. However, considering that the previous male data was acquired via a different cardiomyocyte isolation and culture protocol, we would also want to control those variables prior to direct comparison. Similarly, given the differences we have seen in our previously published

data regarding the response to NR between males and females, as well as the abundance of literature suggested by the reviewer in the former revision comments, we intend to keep this current story focused on female mice and followup with rigorous experiments to compare sex differences in future research.

2/ The authors are honestly showing in supplement Figure 1B and C, that a second run of experience yielded a less robust oscillation in adult female cardiomyocytes. This reviewer understands these experiments that are performed for 36 h on adult mouse cardiomyocytes are difficult and agree that altogether the results essentially corroborate their in vivo data. Never the less, it would be more correct to show in the main figure a pool analysis of the 2 experiences, even if pval is above 0.05, the trend should be there. Or maybe, if the Editor concur, it could be possible to pool results from males and females for more statistical power and then show details in supplement to highlight individual experiences in supplementary figures, showing that the technique is subjected to batch variation. Ideally a 3rd run of experiment on the female cardiomyocytes would be good. Eventually I suggest adding BDM or blebbistatin to the cell culture medium to minimize cell rounding over time if it does not affect too much the response of course.

The question and possible solutions raised by the reviewer were taken in earnest; we have adjusted Figure 1F and Supplementary Figure 1B/C in ways we find more satisfactory and hope that the Reviewer and Editor will agree. As discussed in the comment above, we decided not to combine the male mice since we did not have the corresponding transcriptomic data or other experiments, but did succeed in simultaneously displaying both female experiments on the same graphs and merging them for statistical power. For transparency and clarity, we also included the raw (non-detrended) bioluminescence acquisitions in the supplementary figures for each experiment separately. After normalizing and merging the experiments, we are happy to report that the difference in amplitude between young and old rhythmicity is still statistically significant. Unfortunately, a 3rd run of the experiment was not feasible due to the lack of aged mice, but we believe the replicate wells and duplication of the experiment are sufficiently trending to conclude that aging does effect cardiac rhythmicity. On the reviewer's final point: while our isolation buffers do use BDM to maintain CMs, we did actually find that the addition of BDM to the culture medium impacted our visualization of signal, and was one reason we had to limit the window of recording before significant cell rounding/death occurred.

3/ I also appreciate the effort put by the authors to analyze NMRK2 protein level in the heart and it could be interesting to show it. However, mouse NMRK2 predicted size is 22,375 kDa (Q9D7C9 NRK2_MOUSE in UNIPROT database), hence it is very close to light chain IgG at 25 kDa that can be present in mouse tissues. As the antibody that the authors used is a mouse monoclonal requiring the use of secondary anti-mouse IgG, there is always a risk to reveal IgG. The band shown in gel in the FigC of rebuttal is unfortunately too fuzzy to convincingly demonstrate it is NMRK2. The authors could use gels dedicated to small size proteins to get sharper bands and assess presence or absence of IgG in their protein extract using 2ndary antibody only. However, if this is too difficult because samples are missing, I don't consider this as a key point for the article.

The authors also contend that the NMRK2 Western blot is frustratingly unclear; we have extensively discussed with experts in the field working on this particular protein in search of a solution. From both our own experience and the external advice we have received, the pursuit of the protein is fraught with poor antibodies, unexpected gel migration relative to size, and proximity to interfering native mouse protein, as pointed out by the author. As we lack NMRK2-KO or OE material for controls, as well as running out of appropriately aged, NR-treated material for our own experimental goal here, we unfortunately agree that we cannot confidently demonstrate protein levels of NMRK2 to the degree needed for publication in this manuscript. We also agree with the reviewer that it is not a critical point, as there is still data to support the uptake of NR into cardiac NAD⁺, and even protein level is not a final assessment of enzyme activity. That being said, the authors do believe the subject of NAD dynamics by metabolomics, protein, and enzymatic activity are clear targets to approach and illuminate in future studies.

4/ This reviewer agree that the use of neonate cardiomyocytes that can be cultured over a longer period to study how NAD⁺ modulation affects PER2 rhythmicity is a good idea. And actually, the results are nice but it would seem important to show the level of NAD⁺ reduction.

The authors find this to be a completely valid point; as such, we measured NAD⁺ and NADH levels at two timepoints of the same treatment strategy as presentend in Figure 4. The new Figure 4A/B and text involving it now show that the loss and gain of NAD⁺ via FK866 and NR, respectively, follow the expected outcomes and further strengthen the validation of that model and experiment.

5/ The rationale for switching to HL1 cells after having established the neonate rat cardiomyocytes model is not clear (and not justified in the text). Ex527 works fine in tose cells too and HL1 cells not a very good model of cardiomyocytes. And beyond this, the transcriptomic data do not show the impact of FK866 compared to controls, which seems strange, as well as the lack of pathway enrichment analysis on these data.

The authors thank the Reviewer for pointing out aspects of our experimental design that were not clearly laid out in the manuscript. Considering the need of a new litter of mice for every neonatal cardiomyocyte experiment, and the relative fragility of the primary cells, we opted to focus our efforts on the HL-1 cell line, allowing for much more rapid optimization of treatment duration and concentrations. Similarly, the isolated neonatal mouse cardiomyocytes are relatively poor contributors of RNA in terms of both quality and quantity, limiting their use in our hands to the sensitive imaging and NAD assays. We have changed the discussion text to reflect those decisions. To improve upon and further justify the use of the HL-1 line, we have acknowledged the missing comparisons in our first assessment. We have expanded Figure 5 and relevant text to now incorporate analysis of the FK866 group to controls, as well as the FK866+NR group to controls, representing both the fall to low NAD state and restoration towards the control condition. Likewise, we have compounded this assessment with KEGG analysis in the new Supplemental Figure 4. Our choice of analysis and the results now better mirror the experiments and readings as performed in the *in vivo* cohort.

Minor

1/ *Line 84 and 137: probably reference 15 rather than 19*

The authors commend the Reviewer for their sharp eyes and apologize for our simple error; indeed, those references were mislabelled, and have now been updated as intended.

2/ *For a better global understanding of the data, a short sentence could remind the rationale for performing detrended fluctuation analysis at least in material and methods.*

Exactly as suggested, a few new sentences have been added to the methods section to explain both the process and decision to detrend the adult CM bioluminescence data in Figure 1F; highlight has also been added that the raw data can be found in the supplementary.

3/ *Since the NADH data are available for the cell experiments, it's always better to show it.*

The authors appreciate the recommendation from the Reviewer; we initially did not add our available NADH data since we only had it for cells, but have been convinced by the Reviewer that their addition is still valuable, especially in light of the new results in neonatal CMs. Therefore, we have added all NADH data for the cell-based experiments.

Major Comments for Reviewer 3

"The Authors addressed all points from me and other Reviewers with additional analyses and experiments when possible."

The authors thank the Reviewer for their positive and supportive assessment.

Response to Editor's and Reviewers' Comments

Title: NAD⁺ controls circadian rhythmicity during cardiac aging

Authors: Bryce J. Carpenter, Margaux Lecacheur, Yannick N. Mangold, Kai Cui, Stefan Günther, Marit W. Vermunt, Pieterjan Dierickx

Manuscript ID: COMMSBIO-23-3060-B

General Comments for Reviewer 2

"The authors addressed all points carefully and to the best of their ability. The paper is now very clear and convincing. Congratulation for this excellent study."

The authors thank the Reviewer for their positive and supportive assessment.